# WETAP: Speculative Decoding with Width-Entropy Tree and Adaptive Pruning for LLMs Inference Acceleration

## Abstract

In inference acceleration of Large Language Models (LLMs), speculative decoding is used to coordinate draft model and target model, i.e., sequences are generated at the draft model and then verified in parallel at the target model, where the generation quality and speed of the draft model are the key issues. In this paper, we find that in a token tree, most of the child tokens are grew by few parent tokens with large probabilities in the low-entropy layer, and tokens with small probabilities in deeper layers also have potential to be accepted. Based on these observations, we propose WETAP, first constructing a token tree by determining the width of the next layer based on the entropy of the previous layer, then pruning it by considering both the probability and length of each token to retain the most potential ones. Experiments show that the proposed WETAP improves generation performance by up to 90% and furthermore increases up to 120% speed compared to other SOTA methods.

## 1 Introduction

Nowadays, Large Language Models (LLMs), such as GPT, DeepSeek and LLaMA (Achiam et al., 2023; Liu et al., 2024a; Touvron et al., 2023), have demonstrated powerful reasoning and inference ability, and have extended to every aspect of daily life (Kaplan et al., 2020; Hoffmann et al., 2022; Brown et al., 2020). However, LLMs still have strict usage conditions because of their large-scale weight matrices which present high requirements on devices and cost a lot of energy and time to complete one inference (Dubey et al., 2024). So the inference of LLMs is often impossible to be realized with limited computational power.

In order to decrease the inference cost and latency, speculative decoding (Stern et al., 2018; Xia et al., 2022; Leviathan et al., 2023) incorporates a small draft model which is fast but performance-limited and a large target model which is high-performing but slow to collaboratively complete an inference leveraging their respective advantage. In each iteration, it first generates a draft sequence with $\gamma$ tokens by the draft model *sequentially*, then employs the target model to verify this sequence *in parallel* and sample a new token from the last accepted token's probability distribution. With only one forward pass of the target model and $\gamma$ forward passes of the draft model, it only takes slightly more time than a single forward pass of the large model to generate at least one new token. To further improve the generated length per iteration, in many methods (Miao et al., 2024; Lu et al., 2024), each parent token can grow several child tokens per step to construct a token tree with many layers and many positions (tokens) in one layer. If the number of tokens in the tree exceeds budget, the tree will be pruned based on the probabilities of tokens (Wang et al., 2025; Svirschevski et al., 2024).

However, the existing token tree construction methods simply fix the width of the token tree, ignoring the intrinsic relationship between the adjacent layers in the tree. In general, if tokens within the same layer have more even probabilities, indicating that they are evenly important to grow child tokens, then the width of the next layer should be larger so as more child tokens can be involved in it to improve the acceptance rate because they may have relatively even probabilities to be accepted. In contrast, if tokens within the same layer have more imbalanced probabilities, indicating that tokens with higher probabilities are more important to grow child tokens, we can decrease the width just to

ensure the coverage of child tokens of the more important parent tokens. In addition, the existing methods prune the tree just by probability to approximate token's target probability with the draft one. The approximation works in shallow layers, but it ignores a fact that the degree of correlation between the draft and target models will decrease as layer goes deeper, so retaining tokens with high probabilities in deep layers may not be so helpful. On the contrary, tokens with relatively low probabilities in deep layers may also have potential to be accepted.

Inspired by the above insights, in this paper we propose WETAP which constructs the token tree in a dynamic way, then prunes it adaptively and verifies it from the deep layers to shallow layers. When construction, we determine each layer's width based on the entropy of the previous layer in a proportional relationship. Then we prune it considering the composite score obtained by probability and length. At last, we verify the tree based on cumulative probability of each beam instead of token itself, so we can start verification from the deep to shallow layers and terminate it as long as at least one token is accepted.

Our key contributions are:

- We empirically reveal the relationship between the entropy of the previous layer and growth regularity of the next layer of the token tree in speculative decoding, and the degree of correlation between the draft model and the target model in different layers.

- Building on above, we propose WETAP, a new algorithm that first dynamically constructs a token tree whose width of each layer is determined by the entropy of the previous layer, adaptively prunes it considering both probability and length, and verifies it from the deep to shallow layers.

- Extensive experiments across various models and tasks demonstrate that WETAP consistently improves generated length by up to 90% and furthermore increases up to 120% speed while maintaining the lowest perplexity without any additional module and training, compared to other SOTA methods.

## 2 RELATED WORKS

**Speculative Decoding.** Many methods try to improve acceleration speedup in speculative decoding, for example, (1) substituting the draft model with smaller-scale modules (Li et al., 2024a;b; Cai et al., 2024; Zhang et al., 2024; Cheng et al., 2025), (2) improving speedup by self-speculation (Liu et al., 2024b) or LayerSkip (Xia et al., 2025; Elhoushi et al., 2024), (3) exchanging information like key-value cache or activation between two models to improve alignment (P S et al., 2024; Zimmer et al., 2025). In order to improve speedup while balancing the generation quality, EAGLE (Li et al., 2024b) substitutes the draft model with a trained module which inputs the integration of tokens and features and iterates draft features while drafting. The methods (Zhang et al., 2024; Zimmer et al., 2025) improve quality and speed by introducing on-policy training method or Cross-Attention modules.

**Token Tree Construction.** Many speculative decoding methods construct a draft token tree to be verified by target model. Often the structure of the token tree is predetermined by manually setting the shape or by specifying the width and length in advance (Cai et al., 2024; Luo et al., 2024; Miao et al., 2024; Lu et al., 2024). The manually specified shape often prioritizes ones with high probabilities to extend more beams and selects their child tokens with high probabilities to form the next layer to ensure high acceptance rate. Given specific budget, some methods construct a token tree first larger than budget, then prune it to budget (Wang et al., 2025; Svirschevski et al., 2024).

In contrast, WETAP focuses the intrinsic relationship between adjacent layers in the token tree and the degree of correlation between the draft and target model in different layers, and aims at the sufficient and accurate exploration space of the draft model. WETAP first constructs the token tree with dynamic width and more than budget and then adaptively prunes the tree considering the potential of each token by its probability and length. It is worth noting that WETAP is in no need of any extra module and training which saves extensive time and cost.

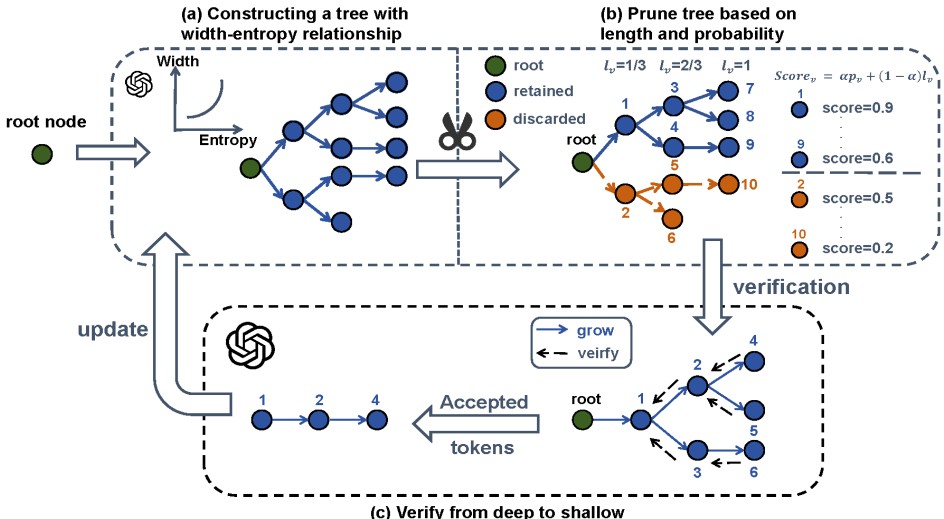

Figure 1: Illustration of our WETAP in one iteration, which includes: (a) Constructing a tree with width-entropy relationship. WETAP determines the width of each layer based on the entropy of the previous layer in a proportional relationship. (b) Pruning the tree based on the length and probability of each token. WETAP considers both probability and length to retain the most potential tokens in the tree. (c) Verifying from deep to shallow layers. WETAP verifies the tree in terms of beam instead of token, so it can verify from deep to shallow layers. If at least one token is accepted, the verification will terminate. Then the generated tokens in this iteration will be updated to the input for the next iteration.

## 3 METHODOLOGY

To improve acceptance rate and speed further, this section first introduces two observations and then WETAP, an algorithm for constructing and reshaping a draft token tree for any input sequence and then verifying it. Algorithm 1 in the Appendix illustrates the pseudo code of WETAP algorithm.

### 3.1 METHODOLOGY OBSERVATIONS

#### 3.1.1 ACCEPTED NUMBER AT DIFFERENT POSITIONS

We investigate the potential of tokens to be accepted at different positions in different layers. For example, we construct a token tree with length of 8 and width of 16, which means the token tree has 8 layers and each layer has 16 positions. If the token has a high probability, it will be at the upper position, and vice versa. Then we study the accepted number of different position 1~16 in layer 1~8 on MT-Bench task as shown in Figure 2. As shown in the left one including all layers and positions, most of accepted tokens are at the upper positions. However as shown in the right one including all layers and last 10 positions, for the lower positions such as 6-16, its accepted number increases as layer goes deeper, which demonstrates that tokens at lower positions of deep layers also have the potential to be accepted. In conclusion, for the same layer, tokens at upper positions are easier to be accepted. For the same position in different layers, tokens with relatively low probabilities are also likely to be accepted in deep layers.

In addition, Figure 2 also demonstrates the degree of correlation between the draft model and the target model. As layer goes deeper, the sum of accepted number of each position will decrease. It can be summarized that the degree of correlation between two models will decrease with layer, so approximating target probability with high draft probability in deep layers may not work effectively.

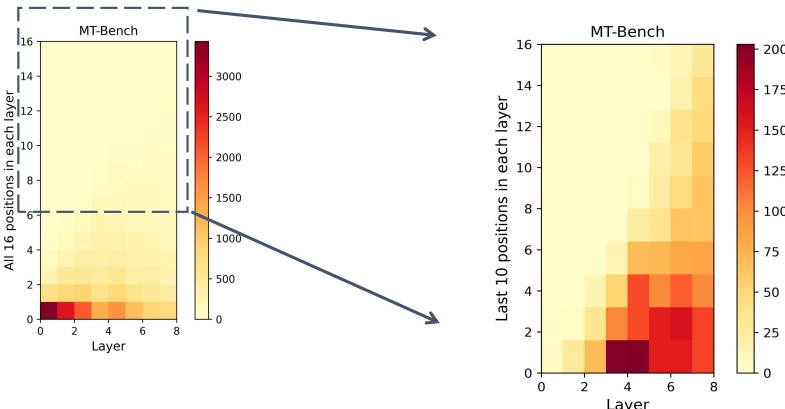

Figure 2: Accepted number of all positions (left) and last 10 positions (right) in different layers.

### 3.1.2 RELATIONSHIP BETWEEN ENTROPY AND GROWTH REGULARITY

Then we explore the correlation between entropy and token's growth regularity. As shown in Fig 3, we can find that when at low entropy, child tokens of tokens with top-3 and top-6 probabilities in the previous layer account for nearly 80% and 90% of tokens in the next layer. It demonstrates that most of child tokens in the next layer are grown by just few parent tokens in the previous layer. However as entropy increases, these proportions decrease. Therefore, it can be summarized that when entropy increases, probabilities of tokens in one layer are distributed more evenly, so every token has relatively even probability to grow child tokens in the next layer, which decrease the proportions of child tokens of highest-ranked ones.

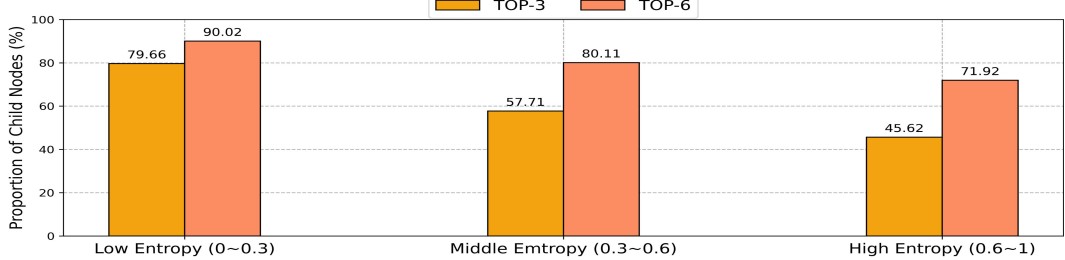

Figure 3: Proportions of child tokens of top 3&6 tokens in the previous layer with low (0∼0.3), middle (0∼0.6) and high (0.6∼1) entropy.

## 3.2 DRAFT AND PRUNE SCHEME

### 3.2.1 CONSTRUCT DYNAMIC-WIDTH TREE

Given predefined budget number of draft tokens, commonly used methods construct a token tree from scratch to the budget number. To improve the generated length, it is necessary to approximate the target probabilities. Many researches (Svirschevski et al., 2024; Wang et al., 2025; Li et al., 2024a) have found the high correlation between draft and target probability. After all, although draft model can not match the performance of target model perfectly, it possesses a certain degree of reasoning ability. So the goal is to construct a token tree $\mathcal{T}$, which has:

$$\max_{\mathcal{T}} \sum_{v \in \mathcal{T}} f(v), \tag{1}$$

where $f(v)$ is the score of each token $v$ in the tree:

$$f(v) = \prod_{\hat{v} \in Parent(v)} p_d(v), \tag{2}$$

where $\hat{v}$ is all parent tokens of $v$ and $p_d(v)$ of the root token is 1, so $f(v)$ is actually the cumulative product probability of the beam for each token. To maximize the sum of each token's cumulative probability, tokens with top probabilities will be selected to form each new layer.

So width of each layer determining the number of selected token is important for the quality of each layer. For example, configuring the width directly by dividing budget number by length restricts the exploration space to a rather small range. In addition, fixed width of each layer fails to consider the intrinsic relationship between adjacent layers. So we can enlarge the width first and configure it specifically in each layer.

Given budget number and length, we can decide the enlarged range of width at first. But width should not be too large because it may incur additional computation and time cost. As shown in Fig 1, then we can specifically determine the width from the range based on the entropy $H_t$ of last layer $t$. Based on the observation in Section 3.1.2, if probabilities of tokens in the previous layer are even, they may have even probabilities to grow child tokens, so the width of next layer should be enlarged to contain more tokens. On the contrary, if probability is entirely concentrated on few tokens, child tokens of them may also have top probabilities, so width can be smaller to just ensure their coverage. So entropy and width can be in a proportional relationship. However, tokens in the same layer may not be from the same distribution, so they must be *P1* normalized first:

$$p'(x_i) = \frac{p(x_i)}{\sum_{j=1}^{W_t} p(x_j)}, \tag{3}$$

where $p(x_i)$ is the original probability and $W_t$ is the width of layer $t$, and then used to calculate the entropy of layer $t$:

$$H_t = -\sum_{i=1}^{W_t} p'(x_i) \log p'(x_i). \tag{4}$$

Then we can determine the width of layer $t+1$ by $H_t$ and predefined width range $W$ using a proportional continuous mapping function:

$$W_{t+1} = W_{\min} + (W_{\max} - W_{\min}) \cdot \left(H_t^{\mathrm{norm}}\right)^\gamma, \tag{5}$$

where $\gamma$ is a continuous coefficient, $W_{\min}$ and $W_{\max}$ is the minimum and maximum width of range $W$ and $H_t^{\mathrm{norm}}$ is the normalized entropy of tokens at layer $t$ ranging from 0 to 1:

$$H_t^{\mathrm{norm}} = \min\left(1, \max\left(0, \frac{H_t}{\log W_t}\right)\right). \tag{6}$$

When $H_t^{\mathrm{norm}} = 0$ and $H_t^{\mathrm{norm}} = 1$, $W_{t+1} = W_{\min}$ and $W_{t+1} = W_{\max}$, respectively.

### 3.2.2 PRUNING BASED ON PROBABILITY AND LENGTH

As mentioned above, we should construct a token tree with the max sum of cumulative product probabilities, so primary criterion for pruning is the probability. Some methods (Wang et al., 2025; Li et al., 2025) also point out the importance of shallow layers' quality to the generated length. However, just considering the probability will retain most of tokens in shallow layers or at the upper positions in a layer and discard others, as the probability of a token in deep layers will be multiplied cumulatively to be very small. But as layers go deeper, the degree of correlation between the draft and target models will decrease, so just retaining tokens with high probabilities in deep layers may not be very effective. So assuming that the tree structure with many beams has ensured the acceptance rate of shallow layers, it is necessary to give more consideration to tokens with relatively low probabilities in deep layers.

To consider tokens at lower positions in deep layers, the beam length can be also introduced as a pruning factor. As shown in Figure 1, the length of a token $l_v$ at the $d$-th layer is normalized to $d/D$, where $D$ is the total length of the tree. To facilitate the weighted calculation of cumulative probability and length, the probability also needs to be normalized to a common range and a comparable magnitude:

$$\frac{p - p_{\min}}{p_{\max} - p_{\min} + \epsilon} \tag{7}$$

where $\epsilon$ is a small constant to avoid division by zero. Then the criterion for pruning is defined as the joint score of each token $v$, combining probability and length:

$$\mathrm{Score}_v = \alpha\, p_v + (1 - \alpha)\, l_v \tag{8}$$

where $\alpha$ is a predefined weight coefficient of length.

However, pruning by probability and length may result in an issue of selecting a token but ignoring its parent tokens. The cumulative probability of a parent token is certainly larger than that of its child, as the latter is derived by further multiplying the parent's cumulative probability by its own probability, so pruning just by probability will not lead to this issue. Therefore, if we prune the token tree by probability and length, we need to supplement the omitted parent tokens first, which will exceed the budget number. So we need to prune it again. We first prune the leaf tokens from shallow to deep layers to minimize the impact on other tokens. If total number still exceeds the budget, then we prune tokenss with lowest probabilities until the number of tokens satisfies the budget. A simple example about pruning the token tree is provided in Appendix D.

### 3.3 Verify Tree from Deep to Shallow Layers

After constructing the token tree, we need to verify it. Target model uses different criteria and verifies the tree token by token and layer by layer. A beam may be discarded because one of tokens is rejected so that the subsequent tokens lose the opportunity to be verified. But research (Qin et al., 2025b) has pointed out that verification can actually be performed on each beam rather than each token so that rejection of preceding tokens will not propagate to subsequent ones with high quality. In summary, we can accept a token based on:

$$\frac{\prod\limits_{\hat{v} \in Parent(v)} p_d(\hat{v})}{\prod\limits_{\hat{v} \in Parent(v)} q_d(\hat{v})} \quad = \quad \prod\limits_{\hat{v} \in Parent(v)} \frac{p_d(\hat{v})}{q_d(\hat{v})} \quad \geq \quad \tau. \tag{9}$$

Previously, we will accept a token if all of its parent tokens and itself are accepted, that is:

$$\frac{p_d(\hat{v})}{q_d(\hat{v})} \quad \geq \quad \tau, \quad \forall \hat{v} \in Parent(v). \tag{10}$$

Now we do not need every token to satisfy this criterion, which will improve the acceptance rate. In addition, these two criteria are not mutually inclusive, it is inappropriate to claim that one criterion is more stringent than the other, so the sequences satisfying these two criteria respectively may not exhibit substantial difference in quality.

Based on the new criterion, we don't need to verify the tree from shallow layers to deep ones. Instead, we can directly verify any token with its parent tokens' probabilities. Therefore, to decrease the time, we propose a new method to start verification from the last layer to shallow ones as shown in Fig 1. If at least one token is accepted, the verification will terminate; otherwise it will continue to the previous layer. Among the accepted tokens in the same layer, the token with most cumulative probability will be selected at last and then target model will sample a new token for the next draft iteration.

## 4 Experiments

### 4.1 Experiment Setups

**Datasets and Models.** In the main paper, we conduct experiments on various text generation tasks to evaluate the effectiveness of our method, including MT-bench (Zheng et al., 2023), HumanEval (Chen et al., 2021) and GSM8K (Cobbe et al., 2021). These tasks and datasets are representative benchmarks for evaluation. We use TinyLLaMA-1.1B as draft model, LLaMA-2-7B and 13B as target models respectively. More details can be found in Appendix B.1 and Appendix B.2.

**Training-free Baselines.** We compare our method with other training-free speculative decoding methods: *SpecInfer* (Miao et al., 2024), *DSBD* (Qin et al., 2025a), *SpecExec* (Svirschevski et al., 2024), *MTAD* (Qin et al., 2025b) and *OPT-Tree* (Wang et al., 2025). All the baselines and our method utilize the same model pairs without any training or fine-tuning. For each method, we generate 1024 new tokens for each input. All methods are stochastic with top-k and top-p sampling with the temperature = 1. More hyper-parameter details can be found in Appendix B.3.

Table 1: Comparison between our and other methods on MT-Bench, HumanEval and GSM8K. We report mean generated length $M$, generation speed $Speed$ (token/s) and perplexity $PPL$.

| Models | Methods | MT-Bench | | | HumanEval | | | GSM8K | | | Mean | | |
|---|---|---|---|---|---|---|---|---|---|---|---|---|---|
| | | $M$ | $Speed$ | $PPL$ | $M$ | $Speed$ | $PPL$ | $M$ | $Speed$ | $PPL$ | $M$ | $Speed$ | $PPL$ |
| LLaMA-2-7B | SPECINFER | 4.20 | 26.95 | 1.22 | 4.87 | 29.03 | 1.23 | 4.02 | 26.04 | 1.39 | 4.36 | 27.34 | 1.28 |
| | DSBD | 3.73 | 22.43 | 1.19 | 4.11 | 24.8 | 1.25 | 3.6 | 21.88 | 1.35 | 3.81 | 23.04 | 1.26 |
| | SPECEXEC | 5.58 | 19.27 | 1.34 | 5.88 | 19.80 | 1.42 | 5.7 | 19.86 | 1.34 | 5.72 | 19.64 | 1.37 |
| | MTAD | 5.62 | 36 | 1.23 | 6.76 | 43.69 | 1.20 | 5.56 | 35.6 | 1.21 | 5.98 | 38.43 | 1.21 |
| | OPT-TREE | 5.43 | 39.95 | 3.49 | 6.34 | 46.12 | 2.21 | 5.56 | 39.24 | 3.08 | 5.78 | 41.77 | 2.93 |
| | WETAP | **7.02** | **42.99** | **1.19** | **7.94** | **48.18** | **1.17** | **7.07** | **43.75** | **1.21** | **7.34** | **44.97** | **1.19** |
| LLaMA-2-13B | SPECINFER | 4.12 | 22.89 | 1.19 | 5.01 | 26.21 | 1.21 | 4.08 | 22.32 | 1.25 | 4.40 | 23.81 | 1.22 |
| | DSBD | 3.56 | 16.12 | 1.19 | 4.37 | 20.55 | 1.21 | 3.59 | 16.96 | 1.24 | 3.84 | 17.88 | 1.21 |
| | SPECEXEC | 5.63 | 17.55 | 1.28 | 5.57 | 17.11 | 1.53 | 5.45 | 17.34 | 1.50 | 5.55 | 17.33 | 1.44 |
| | MTAD | 5.52 | 30.43 | 1.19 | 6.9 | 38 | 1.17 | 5.43 | 30.56 | 1.23 | 5.95 | 33.00 | 1.20 |
| | OPT-TREE | 5.37 | 34.44 | 3.23 | 6.50 | 40.41 | 1.86 | 5.62 | 34.01 | 2.69 | 5.83 | 36.29 | 2.59 |
| | WETAP | **7.03** | **37.80** | **1.18** | **7.96** | **42.43** | **1.14** | **6.98** | **37.32** | **1.19** | **7.32** | **39.18** | **1.17** |

**Evaluation Metrics.** We first report two widely-used metrics for evaluation: mean generated length $M$ and generation speed (token/s). In addition to metrics mentioned above, we then utilize another metric $PPL$ to represent the generation quality and alignment between draft and target models. Detailed descriptions of metrics above can be found in Appendix B.4.

More additional experiments and ablation studies can be found in Appendix C.

## 4.2 MAIN RESULTS

Now we evaluate the effectiveness and efficiency of our method WETAP with other speculative decoding methods. Table 1 presents the comparison between them.

**Effectiveness Analysis.** In terms of effectiveness, WETAP outperforms other methods. It can achieve a mean generated length per iteration of 7.34 for LLaMA-2-7B and 7.29 for LLaMA-2-13B, which is $1.2\times \sim 1.9\times$ across models and tasks. Also it can maintain the lowest perplexity, which is far below *OPT-Tree* and slightly below others. The higher consistency mainly comes from the larger and more accurate draft exploration space and the comprehensive pruning method to retain the most potential tokens. Also, the verification focusing on a beam instead of a single token may accept a sequence which other methods may reject because of part of tokens.

**Efficiency Analysis.** In terms of efficiency, WETAP shows superior efficiency over most of other methods, achieving generation speedup of $1.2\times \sim 2.2\times$, and slightly outperforms *OPT-Tree*. Although the generation speed of WETAP is just marginally higher than *OPT-Tree*, *OPT-Tree* sacrifices quality for faster speed, which may not be optimal in real-world generation where quality and efficiency are both essential. On the contrary, WETAP is more comprehensive simultaneously improving speed and quality. The efficiency advantage of WETAP while not sacrificing quality is driven by the high alignment between draft and target models.

## 4.3 ABLATION STUDY

To provide more insights into two strategies about constructing and pruning the token tree, we conduct the ablation study. We denote WETAP without dynamic width construction as WETAP *w/o dynamic width tree* and WETAP without adaptive pruning as WETAP *w/o prob+len pruning*.

Table 2: Ablation results of speed of WETAP on MT-Bench, HumanEval and GSM8K datasets. (unit: tokens/s)

| Methods | MT-Bench | | HumanEval | | GSM8K | |
|---|---|---|---|---|---|---|
| | LLaMA-2-7B | LLaMA-2-13B | LLaMA-2-7B | LLaMA-2-13B | LLaMA-2-7B | LLaMA-2-13B |
| WETAP *w/o dynamic width tree* | 40.03 | 36.03 | 43.67 | 41.85 | 39.09 | 36.04 |
| WETAP *w/o prob+len pruning* | 40.32 | 36.33 | 47.53 | 41.87 | 42.55 | 34.64 |
| WETAP | **42.99** | **37.80** | **48.18** | **42.43** | **43.75** | **37.32** |

WETAP *w/o dynamic width tree* constructs the token tree with fixed maximum width 128 of the width range and WETAP *w/o prob+len pruning* prunes the token tree just based on probability. We conduct experiments of speed on three tasks with two different target models. As shown in Table 2,

WETAP always achieves the best generation speed. WETAP *w/o dynamic width tree* achieves the worst speedup, which indicates the importance of accurately allocating the construction resources of the tree to where they are needed most. Also the inferior performance of WETAP *w/o prob+len pruning* exhibits the effectiveness of retaining potential tokens in deep layers to improve the speed further.

## 4.4 CASE STUDIES

### 4.4.1 DYNAMIC AND FIXED WIDTH

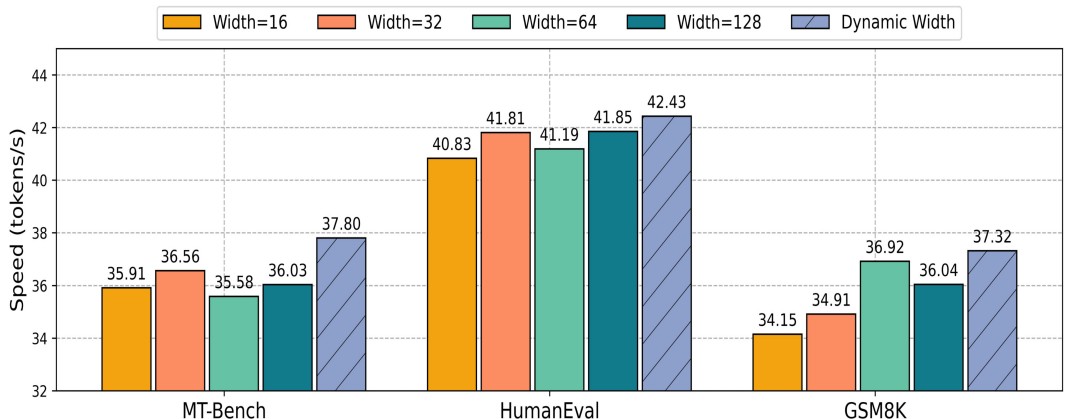

Figure 4: Comparison between fixed and dynamic width with LLaMA-2-13B.

In the main experiments, we set the width range $W \in [16, 128]$ because larger width may instead decrease the speed because of additional cost. So to provide more insights into the width-entropy constructing strategy, we conduct the ablation study about dynamic and fixed width. We select some widths $\{16, 32, 64, 128\}$ from the range including the minimum and the maximum width, and two middle values. As shown in Figure 4, speed will mostly increase with width because it provides larger exploration space to construct a token tree. However the increase is not stable because larger width may incur more computation and time cost. It is worth noting that our method always achieves the best speedup, demonstrating that actually some layers can be accepted with a relatively small width, but some layers may need large ones. So precisely determine the width of each layer based on its condition can save cost while ensuring sufficient and accurate exploration space. Another case study about different width ranges can be found in Appendix C.2.

### 4.4.2 SENSITIVE ANALYSIS OF CONTINUOUS COEFFICIENT

Table 3: $\gamma \in \{0.2, 0.5, 1, 1.2, 2, 4, 8\}$ for different tasks with LLaMA-2-13B. (unit:tokens/s)

| $\gamma$ | MT-Bench | HumanEval | GSM8K |
|---|---|---|---|
| 0.2 | 36.18 | 42.09 | 36.60 |
| 0.5 | 33.54 | 41.55 | **37.32** |
| 1 | 35.34 | 41.53 | 34.80 |
| 1.2 | **37.80** | **42.43** | 35.82 |
| 2 | 36.82 | 41.17 | 34.43 |
| 4 | 34.43 | 38.93 | 36.49 |
| 8 | 33.91 | 41.75 | 35.32 |

We carry out an exploration of continuous coefficient $\gamma$ in the mapping function from entropy to width. As $\gamma$ is the power of $H_t^{norm}$ ranging from 0 to 1, if $\gamma$ is less than 1, $\left(H_t^{\mathrm{norm}}\right)^{\gamma}$ will be larger and so as $W_{t+1}$ to get closer to $W_{max}$ aggressively. On the contrary, if $\gamma$ is more than 1 and becomes larger, $W_{t+1}$ will become smaller to get closer to $W_{min}$ conservatively. So we conduct some experiments to see the effect of different $\gamma$ values $\in \{0.2, 0.5, 1, 1.2, 2, 4, 8\}$ which

represent different degrees of aggressiveness. As shown in Table 3, setting $\gamma$ as 1.2 can achieve the best acceleration for MT-Bench and HumanEval, and 0.5 for GSM8K. The results demonstrate that a relatively balanced mapping from entropy to width is important. Neither too aggressive nor conservative mapping will lead to the performance degradation.

### 4.4.3 SENSITIVE ANALYSIS OF WEIGHT COEFFICIENT

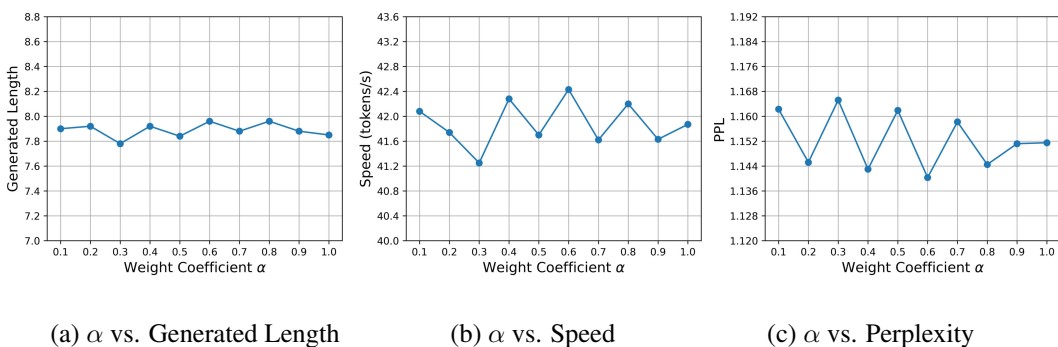

(a) $\alpha$ vs. Generated Length  (b) $\alpha$ vs. Speed  (c) $\alpha$ vs. Perplexity

Figure 5: Performance of WETAP when $\alpha$ ranges from 0 to 1 at intervals of 0.1.

We also investigate the impact of weight coefficient $\alpha$ on the performance of WETAP. Figure 5 presents results of three metrics using LLaMA-2-13B as the target model on HumanEval task when $\alpha$ ranges from 0 to 1 at intervals of 0.1. Three metrics fluctuate as $\alpha$ changes, so we can find that $\alpha$ and performance do not demonstrate a significant (inversely) proportional relationship. However, when $\alpha$ equals 0.6, three metrics all achieve the best performance. It indicates that both probability and length are essential for the evaluation of a token, so the weight coefficient should be balanced to guarantee their joint contributions.

## 5 CONCLUSION

In this work, we introduce WETAP, a dynamic and comprehensive algorithm that first constructs a token tree in a method which establishes a joint consideration of width and entropy between adjacent layers, then prunes it from a more comprehensive perspective to retain the most potential tokens and at last verifies it from deep to shallow layers in terms of beam instead of token. This method can improve performance and efficiency further not requiring any additional module or training. Extensive experiments have demonstrated its superior performance on generated length, speed and perplexity across different models and tasks.

## LLM USAGE STATEMENT

The authors conceived, designed, and carried out this work. A Large Language Model was employed solely for copy-editing the author-written drafts, including grammar, phrasing, and minor formatting.

## ETHICS STATEMENT

All datasets and models we use in the experiments are open-source and publicly released, and no private information is involved in datasets. The scientific observations and experimental results are available with permissive licenses.

## REPRODUCIBILITY STATEMENT

All results reported in this paper are reproducible, and the Supplementary Material contains all necessary codes to replicate them. Experiment setups are discussed in Section 4.1 and Appendix B including the configuration of datasets, models and hyper-parameters.

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

# A    PSEUDO CODE OF WETAP

Here, we give the whole algorithm of our WETAP in detail in Algorithm 1.

---

**Algorithm 1** One Iteration of WETAP Algorithm

---

1: **Input**: draft model $M_q$, target model $M_p$, input prefix $\mathbf{x}$, tree length $L$, width range $W$, budget number $B$, continuous coefficient $\gamma$, weight coefficient $\alpha$, threshold $\tau$.
2:
3: ▷ First construct a token tree $\mathcal{T}$ with dynamic width and length of $L$.
4: $\mathcal{T} \leftarrow \mathbf{x}$                                                   # Initialize the tree with input
5: **for** $i = 1$ to $L$ **do**
6:     **if** i = 1 **then**
7:         $W_i \leftarrow W_{min}$                                  # Width of the first layer is the minimum $W_{min}$
8:     **else**
9:         $W_i \leftarrow \text{GetWidth}(H_{i-1}, W_{i-1}, W, \gamma)$
10:     **end if**
11:     $q_i \leftarrow M_q(\mathbf{x} + [\mathbf{x}_1, \ldots, \mathbf{x}_{i-1}])$
12:     $\mathbf{x}_i, \mathbf{q}_i \leftarrow \text{BeamSample}(q_i, W_i)$           # Select specified number from the distribution
13:     $\mathcal{T} \leftarrow \mathcal{T} + \mathbf{x_i}$
14:     $\mathbf{q}_i^{\text{norm}} \leftarrow \text{Norm}(\mathbf{q}_i)$
15:     $H_i \leftarrow \text{GetEntropy}(\mathbf{q}_i^{\text{norm}})$
16: **end for**
17:
18: ▷ Then prune the token tree to budget number $B$ by probability and length.
19: **if** $|\mathcal{T}| > B$ **then**
20:     **for** $i = 1$ to $L$ **do**
21:         $\mathbf{score}_i = \alpha \mathbf{q_i} + (1 - \alpha)\frac{i}{L}$             # Calculate the composite score of token
22:     **end for**
23:
24:     $\mathcal{T} \leftarrow \text{PruneTree}(\mathcal{T}, \mathbf{score})$
25: **end if**
26:
27: ▷ Last verify the tree from deep layers to shallow ones.
28: $\boldsymbol{P} \leftarrow M_p(\mathcal{T})$
29: $\eta \leftarrow$ -1
30: **for** $i = L$ to $1$ **do**
31:     accept $\leftarrow 0$
32:     **for** $j = 1$ to $|\mathbf{x_i}|$ **do**
33:         ▷ Calculate the cumulative probability of draft and target model.
34:         $p \leftarrow p * \mathbf{P}_{\text{parent}}, q \leftarrow q * \mathbf{q}_{\text{parent}}$
35:         **if** $\min(1, \frac{p}{q}) > \tau$ **then**
36:             accept $\leftarrow$ accept+1
37:             **if** $x_j > \eta$ **then**
38:                 $\eta \leftarrow x_j$
39:             **end if**
40:         **end if**
41:     **end for**
42:     **if** accept $> 0$ **then**
43:         ▷ If at least one token is accepted, terminate the verification.
44:         **Break**
45:     **end if**
46: **end for**
47:
48: $p' \leftarrow P_{\eta+1}$
49: $t \sim p'$
50:
51: **return** $[x_1, \ldots, x_\eta, t]$

---

## B    EXPERIMENTAL SETUPS

### B.1    DATASET CONFIGURATIONS

Our experiments mainly evaluate the effectiveness and efficiency of WETAP on different categories of tasks including multi-round dialogue, mathematical reasoning and code programming. Specifically, we choose MT-Bench (Zheng et al., 2023) which includes diverse open-ended dialogue questions to evaluate conversational ability of models, HumanEval (Chen et al., 2021) which includes hand-written programming problems to evaluate the coding generation and functional correctness, and GSM8K (Cobbe et al., 2021) which includes grade-school math problems to evaluate the arithmetic and reasoning ability. The maximum generation lengths for these three tasks are all 1024. Although the final sequence length is generally less than 1024, we expect models to generate a complete response, similar to how it is expected to perform in the real world.

### B.2    MODEL CONFIGURATIONS

We conduct our experiments on LLaMA-2 series (Touvron et al., 2023) choosing TinyLLaMA-1.1B as draft model and LLaMA-2-7B and LLaMA-2-13B as target models. In our experiments, all models are loaded in the precision of float-16. Our WETAP directly use these models with no need of any additional module, training and fine-tuning.

### B.3    EVALUATION METRICS

We mainly use mean generated length $M$, generation speed *Speed* and perplexity $PPL$ as the evaluation metrics. Specifically, the mean generated length $M$ refers to the average number of output tokens per iteration, including the accepted tokens from the draft model and a new token sampling from the target model at last. Generation speed represents how many tokens are generated per second, reflecting the respective speed of draft and target models and the alignment between them. Last, perplexity $PPL$ is calculated by:

$$PPL(x_{1:n}) = \exp\left(-\frac{1}{n}\sum_{i=1}^{n}\log P_{\text{target}}(x_i \mid x_{<i})\right), \tag{11}$$

where $x_{1:n}$ is the final sequence and $P_{\text{target}}$ is the probability distribution of the target model. Although the final sequence is accepted by the target model, it may not be the best one, so $PPL$ can represent the uncertainty of the target model to the sequence to evaluate the alignment between two models.

### B.4    INFERENCE SETUP

All experiments are conducted in a machine with 1 Nvidia L20 GPU (48GB), 4 CPUs and 100GB main memory. All models are employed on this L20 with enough computational power and memory. For inference, we use batch size of 1, which is more consistent with actual deployment condition and is commonly used in other speculative decoding methods. When comparing with other SOTA methods, we use the same model configuration and device usage for fairness.

For experimental hyper-parameters, we set *top-K* as 10 and *top-P* as 0.9 to select tokens with top probabilities. Then we set the size of the token tree as 64 and length of the tree as 8 for the convenience of calculating the width. For other methods configuring the fixed width, the width of the token tree is 8. For *DSBD* (Qin et al., 2025a), we set the expected number of each layer as 1 to ensure the consistency of verification. For *SpecInfer*, we configure the shape of the tree like a fork where each token must have one and only one child token.

## C    MORE EXPERIMENT RESULTS

### C.1    EVALUATION RESULTS OF VICUNA-7B AND VICUNA-13B

To demonstrate the applicability of WETAP, we provide more evaluation results of Vicuna-7B and Vicuna-13B (Fan et al., 2025). We also compare WETAP with the same SOTA methods in Sec-

Table 4: Comparison between our and other methods on MT-Bench, HumanEval and GSM8K. We report mean generated length $M$, generation speed $Speed$ (token/s) and perplexity $PPL$.

| Models | Methods | MT-Bench | | | HumanEval | | | GSM8K | | | Mean | | |
|---|---|---|---|---|---|---|---|---|---|---|---|---|---|
| | | $M$ | $Speed$ | $PPL$ | $M$ | $Speed$ | $PPL$ | $M$ | $Speed$ | $PPL$ | $M$ | $Speed$ | $PPL$ |
| Vicuna-7B | SPECINFER | 4.57 | 30.38 | 1.30 | 5.64 | 37.23 | 1.32 | 3.35 | 21.73 | 1.87 | 4.52 | 29.78 | 1.50 |
| | DSBD | 3.79 | 23.54 | 1.30 | 4.48 | 27.43 | 1.34 | 2.49 | 15.40 | 1.73 | 3.59 | 22.12 | 1.46 |
| | SPECEXEC | 5.23 | 18.09 | 1.54 | 5.33 | 18.23 | 1.76 | 4.46 | 15.85 | 1.98 | 5.01 | 17.39 | 1.76 |
| | MTAD | 6.21 | 40.62 | 1.29 | 7.57 | 48.20 | 1.20 | 5.66 | 36.66 | 1.39 | 6.48 | 41.83 | 1.29 |
| | OPT-TREE | 4.74 | 40.99 | 4.47 | 4.84 | 40.81 | 4.04 | 3.71 | 35.84 | 7.01 | 4.43 | 39.21 | 5.17 |
| | WETAP | **7.53** | **45.62** | **1.25** | **7.94** | **49.21** | **1.17** | **7.07** | **43.24** | **1.21** | **7.51** | **45.68** | **1.21** |
| Vicuna-13B | SPECINFER | 4.32 | 24.97 | 1.31 | 5.71 | 32.63 | 1.29 | 4.08 | 22.32 | 1.25 | 4.70 | 26.64 | 1.28 |
| | DSBD | 3.65 | 19.60 | 1.29 | 4.61 | 24.41 | 1.30 | 2.90 | 15.63 | 1.56 | 3.72 | 19.88 | 1.38 |
| | SPECEXEC | 5.61 | 16.86 | 1.45 | 5.81 | 17.77 | 1.58 | 4.79 | 15.15 | 1.80 | 5.40 | 16.59 | 1.61 |
| | MTAD | 5.91 | 33.31 | 1.26 | 7.48 | 41.38 | 1.17 | 6.05 | 34.02 | 1.26 | 6.48 | 36.24 | 1.23 |
| | OPT-TREE | 4.91 | 35.45 | 3.23 | 6.50 | 40.41 | 1.86 | 4.26 | 31.56 | 5.48 | 5.22 | 35.81 | 3.52 |
| | WETAP | **7.29** | **38.84** | **1.26** | **8.47** | **45.16** | **1.14** | **6.98** | **37.50** | **1.19** | **7.58** | **40.50** | **1.20** |

tion 4.2 on MT-Bench, HumanEval and GSM8K tasks and evaluate three metrics: mean generated length $M$, generation speed and perplexity $PPL$ of target model to the finale sequence. As shown in Table 4, WETAP achieves the best performance on three metrics. In specific, WETAP achieves the mean generated length of 7.51 for Vicuna-7B and 7.58 for Vicuna-13B, which is $1.2\times \sim 2.1\times$ across models and tasks. In addition, WETAP can also achieve the best speedup by $1.1\times \sim 2.4\times$ compared to other methods. Also the perplexity of WETAP is the lowest among them. In summary, WETAP can also achieve the best generated length, speed and perplexity with Vicuna-7B and Vicuna-13B, demonstrating its applicability to other models.

## C.2 DIFFERENT WIDTH RANGES

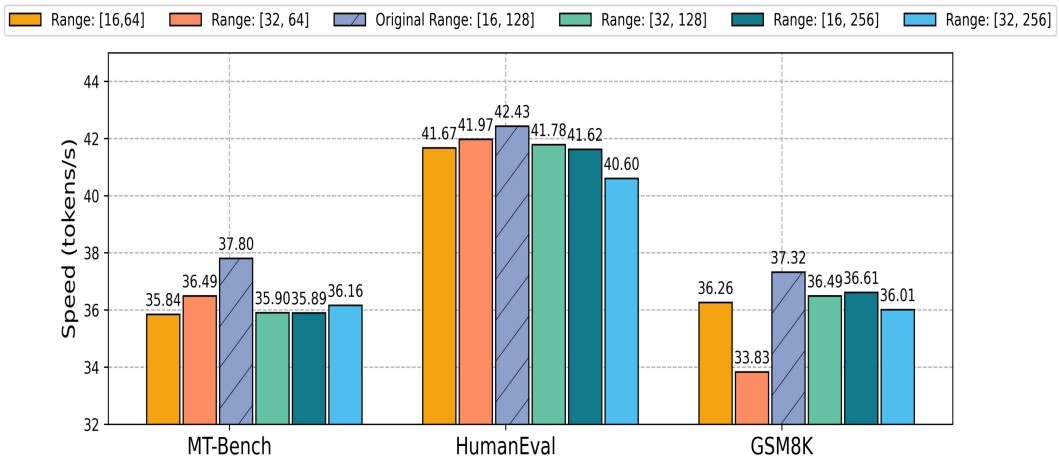

Figure 6: Comparison about different width ranges with LLaMA-2-13B.

In Section 4.2, the width range $W \in [16, 128]$. The results in Section 4.4.1 demonstrate that a proper minimum width can ensure the sufficient exploration space of the token tree, and an appropriate maximum width does not incur additional computational overhead. So in this section, we will further explore the impact of different width ranges to the speed of WETAP. On the basis of the original range, we will change the minimum or the maximum value respectively or simultaneously. So the new ranges include: $[16, 64]$, $[32, 64]$, $[32, 128]$, $[16, 256]$ and $[32, 256]$, and the averages of these ranges show an increasing trend. As shown in Figure 6, actually the speed may not increase as the width range increases. In contrast, a proper width range $[16, 128]$ which is configured in WETAP can achieve the best performance across all tasks. Although a larger width range can determine a larger width for each layer, it can not allocate the construction resources accurately. It enlarges the exploration space by just increasing each layer's width directly, but ignores the specific condition of each layer. On the contrary, a larger width may incur more computational overhead and time cost because the draft model needs to complete the forward pass of all tokens in parallel. So a width

range with proper minimum and maximum value can enlarge the exploration space sufficiently to improve speed while not incurring extra overhead.

### C.3 DRAFT SEQUENCE LENGTH

We investigate the impact of draft sequence length to the generation speed. In addition to the draft sequence length of 8 configured in WETAP, we conduct experiments on other lengths which ranges from $[4, 10]$.

Table 5: Draft Sequence Length for different tasks with LLaMA-2-13B. (unit:tokens/s)

| Length | MT-Bench | HumanEval | GSM8K |
|--------|----------|-----------|-------|
| 4 | 35.35 | 39.48 | 37.08 |
| 5 | 37.17 | 40.26 | 37.23 |
| 6 | 35.40 | 41.71 | 35.23 |
| 7 | 37.07 | 41.53 | 35.65 |
| 8 | **37.80** | **42.43** | **37.32** |
| 9 | 35.95 | 41.72 | 34.78 |
| 10 | 34.98 | 40.14 | 35.33 |

As shown in Table 5, the sequence length of 8 can achieve the best speed performance across all tasks. When sequence length is too small, it takes many iterations to achieve the max generation length or the end token, which may waste extra time. When sequence length is too large, because of the decrease in the degree of correlation between draft and target models, they instead incur extra draft time while not helpful to improve the generated length and speed. So the sequence length of 8 is an appropriate length to ensure sufficient acceptance and not to incur additional overhead.

### C.4 WEIGHT COEFFICIENT FOR MT-BENCH AND GSM8K

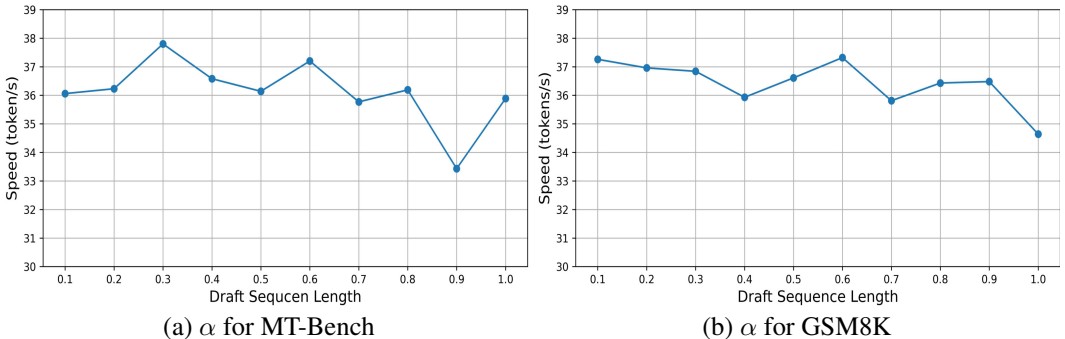

(a) $\alpha$ for MT-Bench         (b) $\alpha$ for GSM8K

Figure 7: Weight Coefficient $\alpha$ for MT-Bench and GSM8K with LLaMA-2-13B.

In Section 4.4.3, we conduct experiments to analyze the effects of different weight coefficient $\alpha$ on HumanEval task. In this section, we further explore the impact of $\alpha$ on MT-Bench and GSM8K tasks. As shown in Figure 7, speed achieves the best when $\alpha$ is 0.3 for MT-Bench and 0.6 for GSM8K. Results both demonstrate a relatively balanced weight coefficient $\alpha$ can lead to the best performance regardless of whether probability or length takes the leading role in the weighted computation.

## D A SIMPLE PRUNING EXAMPLE

In this section, we provide a simple example about pruning a token tree. We configure the length of the tree as 4, width range $W \in [3, 6]$ and the budget number as 12. As shown in the top left corner of Figure 8, we first construct a token tree with dynamic width, where the width of each layer is 3, 5, 4, 4. It is worth noting that the order of tokens in the same layer is not increasing from top

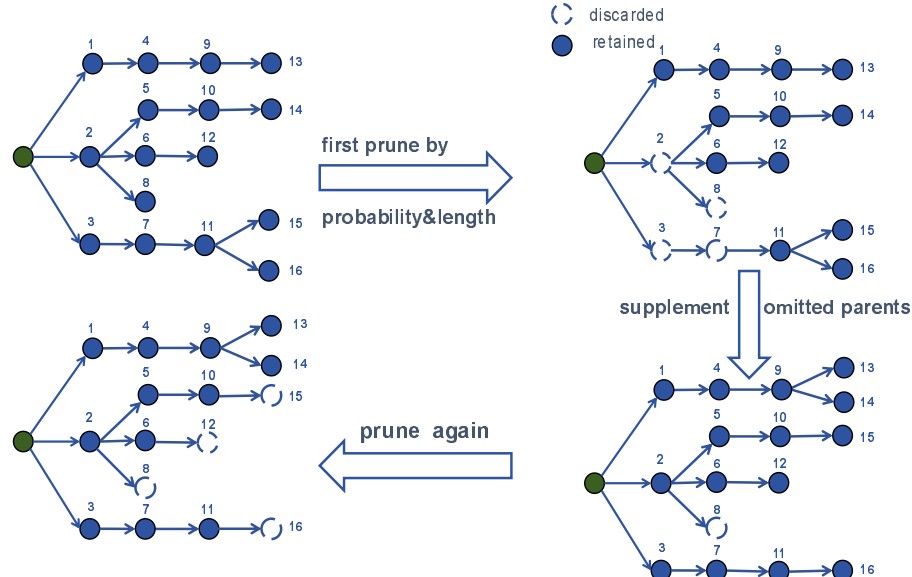

Figure 8: A simple example about pruning a token tree.

to bottom, but rather increasing with respect to their probabilities. For example, token 7 is under token 8 because of the orders of their parent tokens, but the probability of token 7 is higher than the probability of token 8.

Now the number of tokens is 16 which exceeds the budget number 12, so we need to prune the token tree. We first prune the tree to budget as shown in the top right corner. Now all parent tokens 2, 3, 7 of token 5, 6, 11 respectively are omitted, so they need to be supplemented to the tree as shown in the bottom right corner. As the number exceeds the budget again, so we need to prune it again. We first discard leaf tokens from shallow to deep layers such as the token 12. If the number still exceeds the budget, we will discard the tokens with lowest probabilities such as tokens 15 and 16. At last, we get the final token tree as shown in the bottom left corner.

