# OpenReview forum: "WETAP: Speculative Decoding with Width-Entropy Tree and Adaptive Pruning for LLMs Inference Acceleration"
_ICLR.cc/2026/Conference — Submitted to ICLR 2026_

### Official Review · Reviewer_G78s · 2025-10-31

**Soundness:** 3
**Presentation:** 3
**Contribution:** 3
**Rating:** 6
**Confidence:** 3

**Summary:**

The paper proposes WETAP, a training-free speculative decoding method that dynamically constructs a token tree during inference by leveraging the entropy of the previous layer to determine the width of the next layer, followed by an adaptive pruning strategy that jointly considers token probability and depth. Verification is performed from deep to shallow layers, evaluating beams rather than individual tokens to improve acceptance rates. The method requires no additional modules or training and demonstrates consistent improvements over state-of-the-art baselines in terms of generated length (up to +90%), inference speed (up to +120%), and perplexity across multiple models and tasks (e.g., MT-Bench, HumanEval, GSM8K).

**Strengths:**

Strengths：
Dynamic width allocation based on entropy: WETAP adaptively determines the width of each layer in the token tree according to the entropy of the previous layer, enabling more efficient exploration where it is most needed.

Depth-aware adaptive pruning: By jointly considering token probability and depth (length), WETAP preserves potentially valuable low-probability tokens in deeper layers—where draft-target alignment weakens—improving acceptance rates without sacrificing quality.

Unlike conventional token-by-token verification, WETAP verifies entire beams and starts from the deepest layer, allowing early termination and reducing redundant computation while maintaining high generation fidelity.

**Weaknesses:**

Increased per-step computational overhead: WETAP introduces additional inference-time computations—such as per-layer entropy estimation, dynamic width mapping, and composite score calculation for pruning—which may offset part of the speedup gains. However, the paper lacks a detailed breakdown of the time spent on these auxiliary operations versus the actual draft/target model forward passes. A latency or FLOPs analysis would help clarify whether the net acceleration remains favorable under realistic hardware constraints.

Hyperparameter sensitivity: The method introduces new hyperparameters (γ for entropy mapping, α for pruning weight). Although ablation studies are provided, their optimal values vary across tasks (e.g., α = 0.6 for HumanEval but 0.3 for MT-Bench), raising questions about robustness and the need for task-specific tuning in practice.

**Questions:**

On absolute speed reporting: The paper reports speed in tokens/second, which heavily depends on hardware (e.g., L20 GPU in Appendix B.4). Since this metric is not easily reproducible across setups, would the authors consider adding a vanilla autoregressive decoding baseline under identical conditions and reporting relative speedup (e.g., “1.8× faster than baseline”)? This would make the acceleration claims more interpretable and comparable.

On comparison with trained draft-based methods: The paper evaluates WETAP primarily against training-free baselines (e.g., SpecInfer, OPT-Tree) but does not include comparisons with trained speculative decoding frameworks such as EAGLE, EAGLE-2, or Medusa, which also construct token trees using learnable draft heads. Given that these methods can generate high-quality multi-token candidates through training, how does WETAP’s training-free, entropy-guided tree design compare in terms of both speed and generation quality? Would the proposed dynamic width and adaptive pruning strategies still yield advantages when the draft model is specifically trained for speculative decoding?

---

> ### Author Response · Authors · 2025-11-19
>
> We appreciate your effort spent on carefully reviewing our manuscript and your useful feedback. Our point-to-point responses to your comments are given below. Please note that your original comments are shown in framed boxes.
>
> ```text
> Increased per-step computational overhead: WETAP introduces additional inference-time computations—such as per-layer entropy estimation, dynamic width mapping, and composite score calculation for pruning—which may offset part of the speedup gains. However, the paper lacks a detailed breakdown of the time spent on these auxiliary operations versus the actual draft/target model forward passes. A latency or FLOPs analysis would help clarify whether the net acceleration remains favorable under realistic hardware constraints.
> ```
> We conduct experiments to investigate each extra operation's latency and find negligible extra latency brought by them. Below is the detailed explanation.
>
> The additional operations introduced to the draft inference include calculating each layer's entropy and width, and additional operations brought by the new tree pruning method. To investigate their impacts, we conduct two experiments with TinyLLama-1.1B as the draft model. One is our method WETAP including two extra operations mentioned above while the other one uses the traditional construction method without them. Then we investigate the inference time of width calculation, draft forward pass and tree pruning, and investigate the total time needed in draft stage. The results are shown below (the unit is second):
>
> |              | Width | Forward | Pruning | Total Draft |
> |--------------|------:|--------:|--------:|------------:|
> | **WETAP**        | 0.001 | 0.1183  | 0.0083  | 0.1365      |
> | **Traditional**  | 0     | 0.1145  | 0.0072  | 0.1314      |
>
> These results first demonstrate that additional width and entropy calculation introduce almost no additional latency. In addition, theoretically width-entropy operation introduces just *5T+1* FLOPS for a layer with *T* tokens. Because *T* falls in the range of 16-128, so it is negligible compared to every matrix calculation in model forward pass. In addition, WETAP's pruning just takes 0.0011 seconds more than the traditional one, which is also negligible compared to the forward time by the draft model.
>
> ```text
> Hyperparameter sensitivity: The method introduces new hyperparameters (γ for entropy mapping, α for pruning weight). Although ablation studies are provided, their optimal values vary across tasks (e.g., α = 0.6 for HumanEval but 0.3 for MT-Bench), raising questions about robustness and the need for task-specific tuning in practice.
> ```
> It is worth noting that hyperparameters $\alpha$ and $\gamma$ actually are **not so sensitive to datasets**. For the hyperparameter $\gamma$, we find that maintaining 1.2 can almost achieve similar inference speed in different datasets. For $\alpha$, the optimal one may change with datasets. So for the robustness of them, we **apply the best-performing hyperparameter settings of one dataset to others** to examine the changes in speed. The optimal ($\alpha$, $\gamma$) for MT-Bench is (0.3, 1.2), while (0.6, 1.2) for HumanEval and (0.6, 0.5) for GSM8K. The results are shown below:
>
> | ($\alpha$, $\gamma$) |      MT-Bench |     HumanEval |         GSM8K |
> | -------------------- | ------------: | ------------: | ------------: |
> | (0.3, 1.2)           |     **37.80** | 41.25 (-2.8%) | 36.61 (-1.9%) |
> | (0.6, 1.2)           | 37.20 (-1.6%) |     **42.43** | 35.82 (-4.0%) |
> | (0.6, 0.5)           | 35.40 (-6.3%) | 41.55 (-2.1%) |     **37.32** |
>
> The results demonstrate that although using a sub-optimal combination, the speed will not degrade so much, which demonstrates that different combinations can achieve competitive performance and show relatively good robustness.
>
> In addition, it is expected that the optimal settings of these hyperparameters may vary across datasets, as different datasets naturally exhibit different categories and statistical characteristics.

---

> > ### Author Response · Authors · 2025-11-21
> >
> > ```text
> > On absolute speed reporting: The paper reports speed in tokens/second, which heavily depends on hardware (e.g., L20 GPU in Appendix B.4). Since this metric is not easily reproducible across setups, would the authors consider adding a vanilla autoregressive decoding baseline under identical conditions and reporting relative speedup (e.g., “1.8× faster than baseline”)? This would make the acceleration claims more interpretable and comparable.
> > ```
> > To supplement the relative speed report, we conduct the inference experiments by Llama-2-7B and Llama-2-13B respectively on the same three datasets as in the manuscript including MT-Bench, HumanEval and GSM8K. The results are shown below:
> > ### LLaMA-2-7B
> >
> > | Method     | MT-Bench M | MT-Bench Speedup |  HumanEval M | HumanEval Speedup |  GSM8K M | GSM8K Speedup | Overall M | Overall Speed | Overall Speedup  |
> > |------------|------------:|---------------:|--------------:|-------------:|----------------:|---------------:|---------:|-------------:|-----------:|
> > | Vanilla  | 1.00 | 1.00× | 1.00 | 1.00× | 1.00 | 1.00× | 1.00 | 25.39 | 1.00× |
> > | SpecInfer  | 4.20 | 1.06× | 4.87 | 1.15× | 4.02 | 1.02× | 4.36 | 27.34 | 1.08× |
> > | DSBD       | 3.73 | 0.88× | 4.11 | 0.98× | 3.6  | 0.86× | 3.81  | 23.04 | 0.91× |
> > | SpecExec   | 5.58 | 0.76× | 5.88 | 0.78× | 5.7 | 0.78× | 5.72  | 19.64 | 0.77× |
> > | MTAD       | 5.62 | 1.42× | 6.76 | 1.73× | 5.56 | 1.40× | 5.98 | 38.43  | 1.52× |
> > | Opt-tree   | 5.43 | 1.57× | 6.34 | 1.82× | 5.56 | 1.54× | 5.78 | 41.77 | 1.64× |
> > | **WETAP**  | **7.02** | **1.69×** | **7.94** | **1.90×** | **7.07** | **1.72×** | **7.34** | **44.97** | **1.77×** |
> >
> > ### LLaMA-2-13B
> >
> > | Method     | MT-Bench M | MT-Bench Speedup |  HumanEval M | HumanEval Speedup |  GSM8K M | GSM8K Speedup | Overall M | Overall Speed | Overall Speedup  |
> > |------------|------------:|---------------:|--------------:|-------------:|----------------:|---------------:|---------:|-------------:|-----------:|
> > | Vanilla    | 1.00 | 1.00× | 1.00 | 1.00× | 1.00 | 1.00× | 1.00 | 16.64 | 1.00× |
> > | SpecInfer  | 4.12 | 1.38× | 5.01 | 1.58× | 4.08 | 1.33× | 4.40 | 23.81 | 1.43× |
> > | DSBD       | 3.56 | 0.97× | 4.37 | 1.24× | 3.59  | 1.01× | 3.84  | 17.88 | 1.07× |
> > | SpecExec   | 5.63 | 1.06× | 5.57 | 1.03× | 5.45 | 1.03× | 5.55  | 17.33 | 1.04× |
> > | MTAD       | 5.52 | 1.84× | 6.90 | 2.29× | 5.43 | 1.82× | 5.95 | 33.00  | 1.98× |
> > | Opt-tree   | 5.37 | 2.08× | 6.50 | 2.44× | 5.62 | 2.02× | 5.83 | 36.29 | 2.18× |
> > | **WETAP**  | **7.03** | **2.28×** | **7.96** | **2.56×** | **6.98** | **2.22×** | **7.32** | **44.97** | **2.35×** |
> >
> > The results above demonstrate that WETAP outperforms other SOTA methods and can achieve the best speedup in different datasets and models.

---

> > > ### Author Response · Authors · 2025-11-21
> > >
> > > ```text
> > > On comparison with trained draft-based methods: The paper evaluates WETAP primarily against training-free baselines (e.g., SpecInfer, OPT-Tree) but does not include comparisons with trained speculative decoding frameworks such as EAGLE, EAGLE-2, or Medusa, which also construct token trees using learnable draft heads. Given that these methods can generate high-quality multi-token candidates through training, how does WETAP’s training-free, entropy-guided tree design compare in terms of both speed and generation quality? Would the proposed dynamic width and adaptive pruning strategies still yield advantages when the draft model is specifically trained for speculative decoding?
> > > ```
> > >
> > > It is worth noting that actually it is not absolutely fair to compare training-free methods to training-base methods. Our method is explicitly designed for the training-free setting, where model weights cannot be modified and no auxiliary drafter can be trained. So our method is applicable to scenarios where computational resources and time are limited. On the contrary, methods like EAGLE or Medusa assume additional training and optimization, which corresponds to a different problem regime. Thus a direct comparison would not be fully fair, as the assumptions and resource budgets differ substantially.
> > >
> > > In addition, our work focuses on plug-and-play acceleration and how to construct the token tree more dynamically. Training-based methods pursue improvements through learned auxiliary models, representing a different design direction. So it is a promising direction to combine both of them to leverage their respective advantages. If the draft model is trained to align the target model better, theoretically the token tree generated by it can be accepted by the target model more easier. Based on it, the more dynamic and adaptive construction method can retain more potential tokens and further improve the acceptance rate. To investigate the advantages of combination, we conduct experiments on the combination of WETAP and EAGLE on Llama-2-13B when temperature=0. The results are shown below:
> > >
> > > ### LLaMA-2-13B
> > >
> > > | Method     |  MT-Bench Speed |  MT-Bench Speedup | HumanEval Speed | HumanEval Speedup |  GSM8K Speed | GSM8K Speedup | Overall Speed |  Overall Speedup |
> > > |------------|------------:|---------------:|--------------:|-------------:| -------------:| -------------:| -------------:| -------------:|
> > > | Vanilla    | 16.56 | 1.00× | 16.56 | 1.00× | 16.81 | 1.00× | 16.64 | 1.00× |
> > > | EAGLE  | 29.26 | 1.77× | 30.91 | 1.87× | 33.13 | 1.97× | 31.10 | 1.87× |
> > > | **EAGLE+WETAP**  | **32.32** | **2.25×** | **35.87** | **2.17×** | **36.69** | **2.18×** | **34.96** | **2.10×** |
> > >
> > > The results above demonstrate that combination of WETAP and EAGLE can further improve the inference speed and speedup based on EAGLE itself, which show the advantages of combining them to leverage their respective strengths.

---

> > > > ### Author Response · Authors · 2025-11-28
> > > > **Follow-Up: Seeking Further Feedback**
> > > >
> > > > Dear Reviewer G78s, I hope this message finds you well. Since there has been no further response in our recent discussion, we want to kindly follow up and check whether you have any additional questions or concerns regarding our paper and submission above. Your feedback is extremely valuable to us, and we would be more than happy to clarify or elaborate on any remaining points.

---

### Official Review · Reviewer_YL9S · 2025-10-31

**Soundness:** 2
**Presentation:** 3
**Contribution:** 3
**Rating:** 4
**Confidence:** 4

**Summary:**

This paper introduces WETAP, a novel algorithm for Large Language Model inference acceleration via speculative decoding, focusing on dynamically optimizing the draft token tree construction and verification process. WETAP addresses limitations of fixed-width trees by determining each layer's width based on the entropy of the previous layer, thereby efficiently allocating exploration resources based on prediction certainty. It employs an adaptive pruning scheme that calculates a composite score considering both the token's cumulative probability and its depth, ensuring that deep but potentially successful tokens are retained. Furthermore, WETAP implements a "deep-to-shallow" verification strategy that processes candidate beams from the last layer forward, terminating immediately upon acceptance to minimize latency.

**Strengths:**

-  WETAP introduces a principled way to dynamically adjust the width of the draft token tree layers in proportion to the previous layer's output entropy. This mechanism ensures an accurate and sufficient draft exploration space is created, avoiding the computational overhead of overly wide, fixed-structure trees.

-  The method innovatively combines token probability with beam length (depth) in its pruning criterion. This approach effectively addresses the diminishing correlation between draft and target models in deeper layers, ensuring that tokens with relatively lower probabilities but high potential for long acceptance sequences are retained, leading to higher acceptance rates.

**Weaknesses:**

1. Questionable Validity of Deep-to-Shallow Verification Strategy. I do not fully understand the claimed effectiveness of the proposed "deep-to-shallow verification" strategy. On the one hand, the target model's verification process is inherently parallel: all draft tokens in the tree are evaluated via a single forward pass. Therefore, I fail to grasp the fundamental difference or advantage in processing the results sequentially from back-to-front (deep-to-shallow) versus front-to-back (shallow-to-deep). On the other hand, I do not understand why the acceptance of a deep-layer token implicitly guarantees the acceptance of its corresponding shallow-layer parent tokens in the beam. It is a common phenomenon that a misleading token in a shallow layer can cause the deep layers to be "misguided," leading to a draft model output that coincidentally appears similar to the target model's output. In other words, if the deep-to-shallow method can successfully identify an erroneous token upstream, then both front-to-back and deep-to-shallow verification methods should accept the same number of tokens, both halting at the first erroneous draft token.

2. Lack of Temperature=0 Experiments. Although temperature > 0 is a more commonly used strategy, the paper should provide results for  temperature =0 (Greedy Decoding). This is necessary to facilitate smooth reproduction by other researchers and enable a fairer comparison with existing baseline methods.

**Questions:**

- Can your proposed method be combined with existing Token Tree methods? Specifically, if it could be integrated with approaches like EAGLE-3 to further enhance its acceleration capability, the practical utility of WETAP would be significantly improved.

- Your method achieves the lowest PPL in the experimental results. Does this imply that the probability distributions of the generated results from your method's draft model and the target model are the most similar?

- How does your deep-to-shallow verification method guarantee a lossless final acceleration result, particularly in the scenario of Greedy Decoding  (temperature =0) ?

- I recommend that the authors provide performance metrics when integrating this method into practical, production-level libraries like vLLM, as this would greatly demonstrate its real-world deployability and practical value.

If you solve my problems, I'm willing to raise my score.

---

> ### Author Response · Authors · 2025-11-19
>
> We appreciate your effort spent on carefully reviewing our manuscript and your useful feedback. Our point-to-point responses to your comments are given below. Please note that your original comments are shown in framed boxes.
>
> ```text
> Questionable Validity of Deep-to-Shallow Verification Strategy. I do not fully understand the claimed effectiveness of the proposed "deep-to-shallow verification" strategy. On the one hand, the target model's verification process is inherently parallel: all draft tokens in the tree are evaluated via a single forward pass. Therefore, I fail to grasp the fundamental difference or advantage in processing the results sequentially from back-to-front (deep-to-shallow) versus front-to-back (shallow-to-deep). On the other hand, I do not understand why the acceptance of a deep-layer token implicitly guarantees the acceptance of its corresponding shallow-layer parent tokens in the beam. It is a common phenomenon that a misleading token in a shallow layer can cause the deep layers to be "misguided," leading to a draft model output that coincidentally appears similar to the target model's output. In other words, if the deep-to-shallow method can successfully identify an erroneous token upstream, then both front-to-back and deep-to-shallow verification methods should accept the same number of tokens, both halting at the first erroneous draft token.
> ```
>
> You are right that no matter the traversal direction of the token tree, we still need to input all tokens to the target model to get respective target probability. So our opposite traversal direction does not influence the forward pass of the target model while influences the verification progress in the tree.
>
> MTAD[1] propose a new verification strategy which ignores little low-quality tokens in a sequence and accept it only considering the ratio of two probabilities. Based on it, we discover that we can verify the tree from deep to shallow layers because we have already had draft and target probabilities, and this standards can verify every token independently by its two cumulative probabilities ignoring the previous rejected parent tokens.
>
> So the opposite traversal direction can prioritize exploring the longest candidate sequences, enabling the earliest discovery of a feasible sequence, and prioritize eliminating the most unstable candidates, rapidly shrinking the search space and enabling faster failure localization.
>
>
> ```text
> Lack of Temperature=0 Experiments. Although temperature > 0 is a more commonly used strategy, the paper should provide results for temperature =0 (Greedy Decoding). This is necessary to facilitate smooth reproduction by other researchers and enable a fairer comparison with existing baseline methods.
> ```
> Temperature=0 is also a common experimental setup, so we indeed need to investigate our method's performance on it. We conduct experiment when temperature=0 and find that WETAP also outperforms other methods. The results are shown below:
>
> ### LLaMA-2-7B
> | Method     | MT-Bench M | MT-Bench Speedup |  HumanEval M | HumanEval Speedup |  GSM8K M | GSM8K Speedup | Overall M | Overall Speed | Overall Speedup  |
> |------------|------------:|---------------:|--------------:|-------------:|----------------:|---------------:|---------:|-------------:|-----------:|
> | Vanilla  | 1.00 | 1.00× | 1.00 | 1.00× | 1.00 | 1.00× | 1.00 | 25.39 | 1.00× |
> | SpecInfer  | 4.68 | 1.15× | 5.42 | 1.42× | 4.02 | 1.09× | 4.71 | 30.93 | 1.22× |
> | SpecExec   | 5.94 | 0.77× | 6.35 | 0.84× | 6.27 | 0.87× | 6.19  | 20.95 | 0.77× |
> | MTAD       | 5.32 | 1.38× | 6.05 | 1.60× | 5.55 | 1.43× | 5.64 | 37.37  | 1.47× |
> | **WETAP**  | **6.07** | **1.48×** | **6.72** | **1.63×** | **6.37** | **1.53×** | **6.39** | **39.30** | **1.55×** |
>
> ### LLaMA-2-13B
> | Method     | MT-Bench M | MT-Bench Speedup |  HumanEval M | HumanEval Speedup |  GSM8K M | GSM8K Speedup | Overall M | Overall Speed | Overall Speedup  |
> |------------|------------:|---------------:|--------------:|-------------:|----------------:|---------------:|---------:|-------------:|-----------:|
> | Vanilla  | 1.00 | 1.00× | 1.00 | 1.00× | 1.00 | 1.00× | 1.00 | 16.64 | 1.00× |
> | SpecInfer  | 3.8 | 1.36× | 4.85 | 1.68× | 4.1 | 1.47× | 4.25 | 24.96 | 1.50× |
> | SpecExec   | 5.85 | 1.10× | 6.48 | 1.16× | 6.19 | 1.16× | 6.07  | 18.97 | 1.14× |
> | MTAD       | 5.27 | 1.82× | 6.18 | 2.12× | 5.6 | 1.94× | 5.68 | 32.61  | 1.96× |
> | **WETAP**  | **5.96** | **1.97×** | **6.75** | **2.18×** | **6.40** | **2.08×** | **6.37** | **34.55** | **2.08×** |
>
> Temperature=0 (greedy decoding) represents smaller exploration space and verification space, so the generated length and speed of all methods are lower than ones when temperature=1. However, WETAP always outperforms other methods and gets the best speedup among them. This demonstrates the performance of WETAP no matter when temperature=0 or temperature=1.

---

> > ### Author Response · Authors · 2025-11-21
> >
> > ```text
> > Can your proposed method be combined with existing Token Tree methods? Specifically, if it could be integrated with approaches like EAGLE-3 to further enhance its acceleration capability, the practical utility of WETAP would be significantly improved.
> > ```
> > First, in my opinion, it is feasible to combine WETAP with other token tree methods to combine their advantages to construct a better token tree.
> >
> > In addition, it is also a promising research direction to combine WETAP with training-based methods like EAGLE-3. These are two sub-directions within speculative decoding and both dedicated to improving the final outcome through different approaches. One focuses more on providing more potential tokens, while the other one focuses on training a more aligned and light-weighted draft model. So it is theoretically to combine the two to leverage their respective advantages. The more aligned draft model from EAGLE-3 may provide better sequences, and WETAP can further retain more potential ones for verification.
> >
> > To investigate the advantages of the combination, we conduct experiments on the combination of WETAP and EAGLE on Llama-2-13B when temperature=0. The results are shown below:
> >
> > ### LLaMA-2-13B
> >
> > | Method     |  MT-Bench Speed |  MT-Bench Speedup | HumanEval Speed | HumanEval Speedup |  GSM8K Speed | GSM8K Speedup | Overall Speed |  Overall Speedup |
> > |------------|------------:|---------------:|--------------:|-------------:| -------------:| -------------:| -------------:| -------------:|
> > | Vanilla    | 16.56 | 1.00× | 16.56 | 1.00× | 16.81 | 1.00× | 16.64 | 1.00× |
> > | EAGLE  | 29.26 | 1.77× | 30.91 | 1.87× | 33.13 | 1.97× | 31.10 | 1.87× |
> > | **EAGLE+WETAP**  | **32.32** | **2.25×** | **35.87** | **2.17×** | **36.69** | **2.18×** | **34.96** | **2.10×** |
> >
> > The results above demonstrate that combination of WETAP and EAGLE can further improve the inference speed and speedup based on EAGLE itself, which show the advantages of combining them to leverage their respective strengths.
> >
> >
> > ```text
> > Your method achieves the lowest PPL in the experimental results. Does this imply that the probability distributions of the generated results from your method's draft model and the target model are the most similar?
> > ```
> > In my opinion, perplexity and the similarity between two models do not have a direct relationship. Perplexity essentially measures how likely the final sequence is under the large model, and thus reflects the large model’s endorsement of the sequence, rather than the distributional similarity between two models.
> >
> > However, under lossy scenarios, perplexity can serve as an indirect signal of compatibility between the draft and target models, although it should not be viewed as a substitute for more principled distributional metrics such as KL divergence.
> >
> > In summary, given a speculative decoding and verification algorithm, a lower perplexity indicates that the collaboration between two models can, under this algorithm, provide more stable and efficient acceleration while remaining close to the target distribution (or within an acceptable deviation).

---

> > > ### Author Response · Authors · 2025-11-21
> > >
> > > ```text
> > > How does your deep-to-shallow verification method guarantee a lossless final acceleration result, particularly in the scenario of Greedy Decoding (temperature =0) ?
> > > ```
> > > In my opinion, when temperature=0, all speculative decoding methods are lossy no matter which verification method is adopted. Based on Speculative Decoding[2], the draft and target models must sample their tokens by stochastic sampling to ensure each token's sampled probability is the same as the accepted probability. However when using greedy sampling, that is temperature=0, the token with the largest probability must be sampled, which means the sampled probability of it is 1 and it can not be the same as the accepted probability, which destroys the losslessness.
> > >
> > > In addition, our method WETAP also can not ensure the losslessness. Different from traditional verification, MTAD[1] proposes a new verification method mentioned in the first section. Based on it, we further point out that we can start the verification from deep to shallow layers, which is also lossy.
> > >
> > > But through experiments, WETAP can achieve the best speedup and downstream performance compared to other SOTA methods. So, although losslessness is an appealing characteristics in speculative decoding, some lossy methods such as MTAD[1], BiLD[3] and our WETAP can perform better by sacrificing a mild degree of losslessness.
> > >
> > > ```text
> > > I recommend that the authors provide performance metrics when integrating this method into practical, production-level libraries like vLLM, as this would greatly demonstrate its real-world deployability and practical value.
> > > ```
> > > We appreciate the reviewer’s suggestion to integrate our speculative decoding method with vLLM. Conceptually, the two are highly compatible: WETAP accelerates inference at the algorithmic/software level, whereas vLLM is a systems framework, so there is no inherent conflict between them.
> > >
> > > That said, this integration requires substantial systems engineering and optimization work that lies beyond the scope of WETAP and our present research, which focus more on the algorithmic design and improvement of speculative decoding. We view this as a promising combination and are interested to explore such integration in future extensions.
> > >
> > > [1] Optimized multi-token joint decoding with auxiliary model for LLM inference.
> > > [2] Fast Inference from Transformers via Speculative Decoding.
> > > [3] Speculative Decoding with Big Little Decoder.

---

> ### Comment · Reviewer_YL9S · 2025-11-26
>
> Thanks for your detailed response. I appreciate the additional experiments and clarifications, but several of my core concerns remain unresolved:
>
>
> (1) My main concern about the deep-to-shallow verification strategy has not been addressed.
>
> You argue that, under the MTAD-style cumulative probability ratio criterion, you can verify tokens (or paths) from deep to shallow. However, my original objection was about logical justification:
>
> Why does the acceptance of a deeper-layer token (or a deep node in the tree) imply that its  parent tokens are also “reasonable” under the target model?
>
> In your reply, you effectively change the verification rule so that a path can be accepted even if some earlier tokens on that path would have been rejected under a stricter, token-wise criterion. In other words, instead of showing that “deep token acceptance ⇒ parent token should also be acceptable,” you relax the requirement on the parents and move to a path-level, MTAD-style criterion.
>
> This does not resolve the original concern about shallow misleading tokens: a misleading early token may still steer the draft into a region where some deep tokens coincidentally look compatible with the target, and your deep-to-shallow rule can still accept that whole path.
>
>
>
> (2) Since you relax the verification criterion and allow acceptance of paths that would be rejected by strict token-wise verification, your method is, in effect, a loosely speculative decoding method, rather than a strictly lossless/token-wise SPD scheme.
>
> Given this, I am not fully convinced that it is fair to compare WETAP only against “traditional” speculative decoding methods that aim to be (approximately) lossless under their respective definitions. It would be more appropriate to:
>
> Explicitly position WETAP as a loosely-correct speculative decoding algorithm, and compare it against other lossy approaches, such as Judge Decoding[1] and related methods, which also trade a controlled amount of correctness for additional speed.
>
> In addition, for lossy methods, it is not sufficient to report only speedups and final task scores in aggregate. It would be very valuable to explicitly quantify the degradation relative to vanilla decoding (e.g., deltas in MT-Bench, HumanEval, GSM8K vs. the non-accelerated baseline) and to discuss the quality–speed trade-off more systematically.
>
>
> (3) Your response states that, under greedy decoding (temperature = 0), all speculative decoding methods are lossy. I believe this statement is too strong and mixes two different notions of “losslessness”:
>
> Theoretical, distributional losslessness (as in the original speculative decoding paper):
> Here, “lossless” means that, under stochastic sampling, the probability that a token is accepted matches its sampling probability under the target model. In this strict probabilistic sense, your statement about T=0 breaking that guarantee is correct.
>
> Practical, sequence-level losslessness (what I care about in this comment):
> In practice, for greedy decoding, we often care about a different notion:
> If the final generated sequence is exactly the same as what vanilla greedy decoding of the target model would produce, then the method is effectively “lossless” from a functional point of view, regardless of the internal acceptance probabilities.
>
> From this practical perspective, it is not true that all SPD-style methods must be lossy under greedy decoding. In principle, one could design a method that still always reproduces the exact same greedy sequence.
>
> My concern is precisely that your deep-to-shallow verification strategy cannot guarantee this kind of practical losslessness: Because you may accept sequences that differ from vanilla greedy due to the relaxed, path-level criterion. And you do not provide any guarantee (or even a careful analysis) that the final outputs always coincide with target greedy decoding.
>
> [1] Judge Decoding: Faster Speculative Sampling Requires Going Beyond Model Alignment

---

> > ### Author Response · Authors · 2025-11-28
> > **Response to Reviewer YL9S**
> >
> > Thanks for your response.  We are glad  for your appreciation for our additional experiments and clarification. Below, we provide further clarifications and necessary experiments to address your concerns. Please note that your original concerns are shown in framed boxes.
> >
> > ```text
> > Why does the acceptance of a deeper-layer token (or a deep node in the tree) imply that its parent tokens are also “reasonable” under the target model?
> > ```
> >
> > We appreciate your concern about the reasonability of parent tokens by the acceptance of a deep-layer token. We agree that the reasonability of parent tokens can not be strictly guaranteed by our verification method. However, we want to clarify that our method indeed **relaxes** the verification criterion used in lossless schemes, but **the degree of relaxation is controlled and bounded**, rather than unconstrained.
> >
> > More concretely, we accept a sequence based on its cumulative probability of all tokens, not based on the single probability of the final token in it. Even if an early token is locally suboptimal for the target model, the **entire sequence** may still maintain a sufficiently high quality to be accepted. So, although our method allows minor deviations, it **will not accept sequences whose overall probability is clearly implausible** or whose degradation is so substantial caused by misleading shallow tokens.
> >
> > In other words, our method relaxes the strictness of token-level verification, but it does not relax the global quality of the sequence. The verification based on the cumulative probability can effectively suppress the shallow misleading, while enabling a better quality-speedup tradeoff.
> >
> >
> >
> > ```text
> > This does not resolve the original concern about shallow misleading tokens: a misleading early token may still steer the draft into a region where some deep tokens coincidentally look compatible with the target, and your deep-to-shallow rule can still accept that whole path.
> > ```
> >
> > We appreciate your concern about the misleading of the shallow tokens. We agree that the misleading of the shallow tokens may steer the draft into a deviated region. However, we want to clarify that the misleading may not necessarily cause a persistent impact on subsequent token generation.
> >
> > **First**, our verification is based on the cumulative probability of all tokens in a sequence, so actually it  **will not accept sequences that are severely misled by misleading shallow tokens.** Only when the deviation caused by shallow misleading tokens is small enough not to cause a substantial drop in the quality of the whole sequence, it will be accepted.
> >
> > **Second**, when the model generates tokens in an autoregressive way, it puts its attention on all previous tokens to generate subsequent tokens, so even when shallow tokens deviate slightly, the subsequent token distribution may can still naturally steer back toward high-probability semantic region.
> >
> > **Last**, to investigate the **impact of different verification methods on the downstream performance** , we conduct experiments with Llama-2-13b on MT-Bench and HumanEval, using **vanilla target model, lossless methods** like SpecInfer and SpecExec, and **lossy methods** like MTAD and our WETAP. We evaluate the downstream performance on MT-Bench by the score from deepseek-r1 as the judge model, and evaluate the downstream performance on HumanEval by pass@1. Results are shown below:
> >
> > | Metric               | MT-Bench Score | HumanEval pass@1 |
> > | -------------------- | -------------: | ---------------: |
> > | Vanilla Llama-2-13b  |          3.775 |             18.7 |
> > | SpecExec (lossless)  |          5.862 |             15.2 |
> > | SpecInfer (lossless) |          5.875 |             15.2 |
> > | MTAD (lossy)         |          5.825 |             16.5 |
> > | **WETAP (lossy)**    |      **5.975** |         **18.9** |
> >
> > The results above demonstrate that  the lossy methods like MTAD and our WETAP can get similar or even better scores than lossless methods or the vanilla target model. It also demonstrates that our WETAP can generate high quality answers with an appropriate degree of relaxation that enables a larger and more flexible exploration space. So the constrained relaxation and some misleading shallow tokens may not cause a substantial degradation of performance, on the contrary, it may provide more dynamic and diverse final answers.

---

> > > ### Author Response · Authors · 2025-11-28
> > > **Response to Reviewer YL9S**
> > >
> > > ```text
> > > Given this, I am not fully convinced that it is fair to compare WETAP only against “traditional” speculative decoding methods that aim to be (approximately) lossless under their respective definitions. It would be more appropriate to:
> > >
> > > Explicitly position WETAP as a loosely-correct speculative decoding algorithm, and compare it against other lossy approaches, such as Judge Decoding[1] and related methods, which also trade a controlled amount of correctness for additional speed.
> > >
> > > In addition, for lossy methods, it is not sufficient to report only speedups and final task scores in aggregate. It would be very valuable to explicitly quantify the degradation relative to vanilla decoding (e.g., deltas in MT-Bench, HumanEval, GSM8K vs. the non-accelerated baseline) and to discuss the quality–speed trade-off more systematically.
> > > ```
> > > We sincerely appreciate your insightful suggestion to explicitly position WETAP as a loosely-correct speculative decoding method and to compare it against other lossy approaches such as Judge Decoding. This is a valuable recommendation and we sincerely apologize for not including these experiments in the above submission. We also appreciate your emphasis on reporting performance deltas relative to vanilla decoding.
> > >
> > > We want to clarify that Judge Decoding[1] does not release its training tuples including correct and wrong answers, and its model weights or training code, and this method requires substantial time to train an additional linear head on top of target embeddings. Therefore, we compare our WETAP with three other loose-correct or lossy approaches. One is MTAD[2] which we have compared with before. One is BiLD[3] whose verification is based on the **cross entropy of the draft and target probabilities.** The last one is FSD[4] whose verification is based on **Jensen–Shannon (JS) Divergence.**  Like the experiments shown in the previous section, we also conduct experiments on **these four loose-correct methods and vanilla Llama-2-13b model** to investigate their **speed and downstream performance**. The results are shown below:
> > >
> > > | Metric              | MT-Bench Speed (token/s) |   MT-Bench Score | HumanEval Speed (token/s) | HumanEval pass@1 |
> > > | ------------------- | -----------------------: | ---------------: | ------------------------- | ---------------- |
> > > | Vanilla Llama-2-13b |                    16.56 |            3.775 | 16.56                     | 18.7             |
> > > | MTAD                |            30.43 (1.84×) |    5.825 (2.05↑) | 38.00 (2.29×)             | 16.5 (2.2↓)      |
> > > | BiLD                |            35.74 (2.16×) |     4.275 (0.5↑) | 37.43 (2.26×)             | 12.2 (6.5↓)      |
> > > | FSD                 |            20.57 (1.24×) |   5.888 (2.113↑) | 35.66 (2.15×)             | 18.3 (0.4↓)      |
> > > | **WETAP**           |        **37.80 (2.28×)** | **5.975 (2.2↑)** | **42.43 (2.56×)**         | **18.9 (0.2↑)**  |
> > >
> > > Compared to **other lossy methods**, our WETAP outperforms them in terms of both inference speed and downstream performance. Compared to **vanilla Llama-2-13b target model**, WETAP can even outperform it on the downstream performance. **For the dialogue task MT-Bench**, it does not have the single ground-truth answer, so the score primarily depends on the relevance of the final answer to the question as well as its grammaticality and logical coherence. So our WETAP can outperform vanilla target model because it includes an extra draft model to enhance expression and enables a larger and more flexible exploration space to generate more comprehensive answers. **For the coding task HumanEval**, WETAP outperforms vanilla target model due to the normal fluctuations in one or two more correct answers. **To sum up,** the appropriately loose verification can improve the speed while incurring only limited quality degradation, and in some cases may even lead to quality gains due to more dynamic answers.

---

> > > > ### Author Response · Authors · 2025-11-28
> > > > **Response to Reviewer YL9S**
> > > >
> > > > ```text
> > > > My concern is precisely that your deep-to-shallow verification strategy cannot guarantee this kind of practical losslessness: Because you may accept sequences that differ from vanilla greedy due to the relaxed, path-level criterion. And you do not provide any guarantee (or even a careful analysis) that the final outputs always coincide with target greedy decoding.
> > > > ```
> > > > We sincerely appreciate your clarification regarding the losslessness at *T=0* and thank you for pointing out the confusion in our original description. We also appreciate your raising up the concern about the losslessness when greedy decoding. Through your clarification, we now have a deeper understanding of the losslessness at *T=0* and we want to clarify that **our WETAP can guarantee the losslessness at *T=0***. We will explain our losslessness with a specific example below.
> > > >
> > > > Assuming that we now have a draft token tree shown below:
> > > >
> > > > ```text
> > > > Draft Token Tree
> > > >
> > > > O(prefix)
> > > > ├─ A
> > > > │  └─ C
> > > > └─ B
> > > >    └─ D
> > > > ```
> > > >
> > > > The token tree has two layers where the prefix token *O* has two child tokens *A* and *B*, and they have one child token *C* and *D* respectively. Now we input the token tree into the target model and greedily sample token based on each draft token:
> > > >
> > > > ```markdown
> > > > Target model greedily samples new token based on each draft token and output its corresponding target child token. For example, the target child token of token A is A' and its target probability p(A') must be 1.
> > > >
> > > > O -->  O' (p=1)
> > > > A -->  A' (p=1)
> > > > B -->  B' (p=1)
> > > > C -->  C' (p=1)
> > > > D -->  D' (p=1)
> > > > ```
> > > >
> > > > Now we can start the deep-to-shallow verification based on the draft token tree and target tokens. In specific, we **start from the deepest token *C*** and verify its corresponding sequence ***O -->A -->C***.  We verify this sequence by its draft cumulative probability **q(A) · q(C)** and its target cumulative probability **p(A) · p(C)**(The probability of the prefix token *O* is 1). In the following we will clarify the verification:
> > > >
> > > > ```text
> > > > We start the verification of O -->A -->C from the deepest token C.
> > > >
> > > > if C == A':
> > > > 	p(C) = p(A') = 1  -->  continue the verification to the shallower token A
> > > > else:
> > > > 	p(C) = 0  -->  p(A) · p(C) = 0, reject this sequence
> > > >
> > > > Now we assume that token C = token A', so we continue the verification to the shallower token A.
> > > >
> > > > if A == O':
> > > > 	p(A) = p(O') = 1  -->  now p(A) · p(C) = p(O') · p(A') = 1, accept this sequence
> > > > else:
> > > > 	p(A) = 0  -->  p(A) · p(C) = 0, reject this sequence
> > > > ```
> > > >
> > > > As shown above, we will accept the draft sequence ***O -->A -->C*** only when ***A=O'*** and ***C=A'*** , where the final accepted sequence is the same as the target sequence greedily sampled by the target model. So the deep-to-shallow verification can still guarantee the losslessness when *T=0*.
> > > >
> > > > ---
> > > >
> > > > We hope the above clarifications and additional experiments can comprehensively address your concerns. We deeply appreciate your concerns and insightful suggestions. We will incorporate these discussions into our manuscript and remain open to further feedback. Please feel free to reach out with any additional concerns.
> > > >
> > > > Thank you once again sincerely for the time and efforts you have dedicated to reviewing our submission.
> > > >
> > > >
> > > >
> > > > [1] Judge Decoding: Faster Speculative Sampling Requires Going Beyond Model Alignment.
> > > >
> > > > [2] Optimized multi-token joint decoding with auxiliary model for LLM inference.
> > > >
> > > > [3] Speculative Decoding with Big Little Decoder.
> > > >
> > > > [4] Fuzzy Speculative Decoding for a Tunable Accuracy-Runtime Tradeoff.

---

### Official Review · Reviewer_Nves · 2025-11-02

**Soundness:** 3
**Presentation:** 3
**Contribution:** 3
**Rating:** 6
**Confidence:** 4

**Summary:**

This paper improves speculative decoding of LLM based on two observations in a token tree:

1) Most of the child tokens are grew by few parent tokens with large probabilities in the low-entropy layer.
2) The degree of correlation between the draft model and the target model decreases in deeper layers.

Correspondingly, this paper constructs a token tree by determining the width of the next layer based on the entropy of the previous layer, then pruning it by considering both the probability and length of each token. Experiments show that the proposed method improves generation performance and speed compared to SOTAs.

**Strengths:**

1. The key idea of proposed method is clearly represented and well motivated. It is based on two observations about the distributions of accepted tokens w.r.t entropy and layer. Those observations have not been discussed by previous literature.
2. The effectiveness of proposed method is supported by various experiments. It not only improves the speed measured by tokens per second but also improves the generation quality measured by perplexity score.

**Weaknesses:**

1. typos:

a) line 278, "tokenss" -> "tokens"

b) right side of figure 2, the range of Y calibration should be 6 to 16, instead of 0 to 16.

2. The implement details of tree pruning are not clearly represented. It is better to add the pseudo code of tree pruning and analyze its complexity. To maintain the tree structure, it seems that we can only prune leaf nodes.

3. The authors claim that it requires less time to verify tree from deep to shallow layers, compared with shallow to deep layers. However, there are no experiments or further analysis. In my opinion, to obtain the cumulative probability, we must traverse the tree from shallow to deep layers.

**Questions:**

The hyper-parameters are tuned per model/per dataset, or kept the same for all experiments?

---

> ### Author Response · Authors · 2025-11-19
>
> We appreciate your effort spent on carefully reviewing our manuscript and your useful feedback. Our point-to-point responses to your comments are given below. Please note that your original comments are shown in framed boxes.
> ```text
> 1. typos:
> a) line 278, "tokenss" -> "tokens"
>
> b) right side of figure 2, the range of Y calibration should be 6 to 16, instead of 0 to 16.
> ```
> We sincerely appreciate your pointing out the typographical and labeling errors in our manuscript. We are so sorry to make these careless errors and will check our manuscript more carefully to avoid errors again.
>
> ```text
> 2. The implement details of tree pruning are not clearly represented. It is better to add the pseudo code of tree pruning and analyze its complexity. To maintain the tree structure, it seems that we can only prune leaf nodes.
> ```
>
> Regarding the complexity of pruning the token tree, we first assume that the token tree has *N* tokens and *L* layers, and we want to prune it to *W* tokens at last. We need to prune the tree first based on probability and length whose complexity is $O(N \log N)$. Then we need to supplement omitted parent tokens whose complexity is $O(N)$. Now the number of tokens in the tree may be more than *W*, so we need to prune it again by discarding leaf tokens from shallow to deep layers whose complexity is $O(N)$. If the number is still more than *N*, we need to prune it at last based on probability whose complexity is $O(N \log N)$. In summary, the total complexity is $O(N \log N)$, which is the same as the complexity of traditional pruning method just based on probability. So our pruning method can retain more potential tokens without extra complexity.
>
> Then we provide the pseudo code of pruning the token tree below and will add it to the manuscript later.
>
> Let the scoring function be defined as:
> $$
> \text{score} = \alpha \ p + (1-\alpha) \ l
> $$
>
> ```text
> Algorithm Tree-Pruning
> Input: token tree T, width limit W, tree size |T|=N
> Output: pruned token tree T'
>
> 1:  Calculate each token's score by function score(i)
> 2:  Pruning the tree based on score
> 3:  Supplement omitted parent tokens to tree to get T''
> 4:  If |T''| > W:
> 5:      discard leaf tokens from the first to second-to-last layer to get tree T'''
> 6:      T'' = T'''
> 7:      If |T'''| > W:
> 8:          Prune T''' based on probability to get T''''
> 9:          T'' = T''''
> 10:      End if
> 11:  End if
> 12:  T' = T''
> 13:  return the pruned tree T'
> ```
>
> ```text
> 3. The authors claim that it requires less time to verify tree from deep to shallow layers, compared with shallow to deep layers. However, there are no experiments or further analysis. In my opinion, to obtain the cumulative probability, we must traverse the tree from shallow to deep layers.
> ```
>
> First, regarding your question about the **traversal direction of the token tree** to get the probability, we want to explain that when constructing the token tree, we can get each token's cumulative probability based on draft probability. When verification, we input the whole token tree into the target model to get target probability. With these two probabilities, we can start comparing them to verify tokens, so the traversal direction do not affect the probability calculation. And based on MTAD[1], we verify a token independently by its two cumulative probabilities, so we discover that we do not need to verify the token tree layer by layer from shallow to deep. On the contrary, we can traverse the direction from deep to shallow.
>
> Second, regarding the time of different traversal direction, we can reduce the verification time because we can find the longest accepted sequence faster by verifying from deep to shallow layers instead of finding it layer by layer. However, this reduction is rather limited compared to the forward time by target model because we still need to input the whole token tree into the target model. But this slight reduction may still be beneficial in real-time inference scenarios. In addition, changing the traversal direction can enable early elimination of the most unstable sequences in the last layer, thereby quickly shrinking the search space and allowing for faster identification of failure ones.

---

> ### Author Response · Authors · 2025-11-19
>
> ```text
> The hyper-parameters are tuned per model/per dataset, or kept the same for all experiments?
> ```
> It is common for deep learning models and methods to exhibit a certain degree of hyperparameter sensitivity across datasets, as different datasets naturally exhibit diverse categories and statistical characteristics. However, in our WETAP, the **hyperparameters $\alpha$ and $\gamma$ are not particularly sensitive to datasets** in practice.
>
> To assess robustness of them more concretely, we conduct a **transfer experiment: we apply the best-performing hyperparameter setting of one dataset to others** to examine the changes in inference speed. It is worth noting that the optimal ($\alpha$, $\gamma$) for MT-Bench is (0.3, 1.2), while (0.6, 1.2) for HumanEval and (0.6, 0.5) for GSM8K. The results are shown below:
>
> | ($\alpha$, $\gamma$) |      MT-Bench |     HumanEval |         GSM8K |
> | -------------------- | ------------: | ------------: | ------------: |
> | (0.3, 1.2)           |     **37.80** | 41.25 (-2.8%) | 36.61 (-1.9%) |
> | (0.6, 1.2)           | 37.20 (-1.6%) |     **42.43** | 35.82 (-4.0%) |
> | (0.6, 0.5)           | 35.40 (-6.3%) | 41.55 (-2.1%) |     **37.32** |
>
> The results show that although using a sub-optimal hyperparameter settings, WETAP **exhibits strong stability**: five out of six transferred settings show less than 4% degradation, and even the worst case remains within a modest 7% drop. More importantly, WETAP using the sub-optimal  hyperparameter settings can still **outperform** other methods.
>
> Taken together, these results indicate that WETAP can maintain competitive performance across different hyperparameter settings and demonstrate relatively strong robustness.
>
> [1] Optimized multi-token joint decoding with auxiliary model for LLM inference.

---

> > ### Author Response · Authors · 2025-11-28
> > **Follow-Up: Seeking Further Feedback**
> >
> > Dear Reviewer Nves, I hope this message finds you well. Since there has been no further response in our recent discussion, we wanted to kindly follow up and check whether you have any additional questions or concerns regarding our paper and submission above. Your feedback is extremely valuable to us, and we would be more than happy to clarify or elaborate on any remaining points.

---

### Official Review · Reviewer_BxfY · 2025-11-04

**Soundness:** 2
**Presentation:** 3
**Contribution:** 3
**Rating:** 2
**Confidence:** 4

**Summary:**

This paper proposes extensions to popular tree-based speculative decoding methods for enhancing the speedup. The improvements include: (i) dynamically constructing the token tree by adaptively changing the width of each layer; (ii) pruning the token trees based on length as well probabilities of the tokens in each layer; (iii) verifying the tokens from right to left instead of left to right to increase the chances of acceptance.

**Strengths:**

- The speedup results are compelling -- the proposed method seems to be faster than comparable baselines.
- The core idea of constructing the token tree dynamically, based on the observed probabilities of tokens in each layer makes sense, and is empirically validated.
- There is a sufficiently detailed ablation analysis at the end which shows the impact of the hyperparameters introduced.

**Weaknesses:**

- When verifying from deep to shallow layers, it is not clear if the procedure still preserves the guarantees from speculative decoding that the resulting distribution will exactly match the target distribution. If this is not the case, there needs to be a more rigorous evaluation of any accuracy trade-offs that the method makes.
- On a related note, I find the use of perplexity as an eval metric strange here. Perplexity is generally measured against ground truth data, whereas here it is being used on the model generations itself. Hence, it is an uncertainty /entropy metric rather than telling us something about the quality of the resulting model. Combined with the fact that it is not clear if the SD guarantee holds, it is important to verify the model performance using actual downstream task metrics.
- Comparing the results in tables 1 and 2, it seems that most of speedup actually does come from the modified verification strategy -- removing the dynamic tree construction or the pruning strategy has little impact compared to the where the baselines are. Again, we need more discussion about what the modified verification strategy is actually doing to the output distribution.
- The method seems somewhat sensitive to the choice of hyperparameters -- so it would be good to see a generalization study where the gamma and alpha are tuned on one dataset and used on another one.

**Questions:**

See weaknesses.

---

> ### Author Response · Authors · 2025-11-19
>
> We appreciate your effort spent on carefully reviewing our manuscript and your useful feedback. Our point-to-point responses to your comments are given below. Please note that your original comments are shown in framed boxes.
>
> ```text
> When verifying from deep to shallow layers, it is not clear if the procedure still preserves the guarantees from speculative decoding that the resulting distribution will exactly match the target distribution. If this is not the case, there needs to be a more rigorous evaluation of any accuracy trade-offs that the method makes.
> ```
> You are right that our verification method does not preserve the distributional consistency from Speculative Decoding[1]. It is worth noting that we propose a new verification direction from deep to shallow based on the verification strategy proposed by MTAD[2]. Different from the traditional method, MTAD[2] directly accepts a sequence if its likelihood ratio satisfies $$
> \frac{p(x_{t+1:t+i} \mid x_{1:t})}{q(x_{t+1:t+i} \mid x_{1:t})} \ge \tau.
> $$
> This verification method may ignore little low-quality tokens in a sequence and accept it only considering the ratio of two cumulative probability. Based on it, we discover that we can verify the tree from deep to shallow layers because we have already had draft and target probabilities, and we can verify every token independently by its two cumulative probabilities. So it destroys the distributional consistency proposed by Speculative Decoding[1].
>
> However, on one hand, MTAD[1] has provided a theoretical bound on the approximation error between MTAD and MTJD. On the other hand, different verification strategies may have different design objectives. Lossless methods aim at distributional consistency, while lossy methods such as MTAD[2], BiLD[3] and our WETAP may aim at higher speedup and better downstream performance sacrificing a mild degree of losslessness. The results about downstream performance can be found in the next section.
>
> In our experiment results, we have demonstrated a considerable speedup. In addition, we want to use the metric **perplexity** to partly evaluate the degree of lossiness by partly investigating the degree of distributional deviation induced by the lossy process. And our method can always achieve the lowest perplexity compared to other SOTA methods. But we indeed lack an evaluation of how the lossiness affects **performance on specific downstream tasks**, so we will provide it in the following.
>
> ```text
> On a related note, I find the use of perplexity as an eval metric strange here. Perplexity is generally measured against ground truth data, whereas here it is being used on the model generations itself. Hence, it is an uncertainty /entropy metric rather than telling us something about the quality of the resulting model. Combined with the fact that it is not clear if the SD guarantee holds, it is important to verify the model performance using actual downstream task metrics.
> ```
>
> First we want to explain that we use the most common metrics: **generated length** ***M*** and **inference speed** like other speculative decoding methods such as MTAD[2], BiLD[3] and SWIFT[4]. And in the following, we will provide more experiments about the **downstream performance** of different methods and find that WETAP also outperforms others.
>
> In addition, as mentioned above, we want to use **perplexity** to serve as a proxy for the degree of lossiness since a lossy decoding method generally yields sequences that the target model assigns lower probability to.
>
> However, perplexity reflects only the sequence-level probability deviation, not the full distributional deviation. So it is necessary to evaluate the influence of lossiness by downstream performance. We conduct experiments on datasets MT-Bench and HumanEval using Tiny-Llama-1.1B as the draft model and Llama-2-13B as the target model. It is worth noting that we do not evaluate the downstream performance on GSM8K dataset because the final performance of speculative decoding is ultimately constrained by the target model. The results are shown below:
>
> |              | MT-Bench score | HumanEval pass@1 |
> |--------------|---------------:|-----------------:|
> | Vanilla | 3.775 | 18.7  |
> | SpecExec | 5.862 | 15.2  |
> | SpecInfer | 5.875 | 15.2  |
> | MTAD  | 5.825     | 16.5  |
> | WETAP | 5.975     | 18.9  |
>
> The results above demonstrate that the lossless and lossy methods like MTAD and WETAP can all achieve similar downstream performance. We also find that WETAP can achieve the best downstream performance compared to other methods and even the vanilla Llama2-13B. So it can be summarized that lossy methods may not degrade the downstream performance. On the contrary, a mild degree of lossiness or a new verification method can adapt the verification standards and generate potential sequences finally.

---

> ### Author Response · Authors · 2025-11-19
>
> ```text
> Comparing the results in tables 1 and 2, it seems that most of speedup actually does come from the modified verification strategy -- removing the dynamic tree construction or the pruning strategy has little impact compared to the where the baselines are. Again, we need more discussion about what the modified verification strategy is actually doing to the output distribution.
> ```
> It is worth noting that Table2 in our manuscript only conducts the ablation experiments on entropy-width construction and tree pruning separately while does not regard these two components above as an integrated whole to construct the tree and conducts ablation experiments on tree construction and verification. So it may not be clear to investigate gain from each component in WETAP.
>
> In the following, we conduct ablation experiments on tree construction and verification. Below is the experimental details, which demonstrate that tree construction contributes more to the speedup.
>
> To conduct experiments to investigate gain from different parts, we first divide WETAP into two parts: token tree construction which includes entropy-width relationship and the new tree pruning method, and the deep-to-shallow verification. The experiments start from a baseline without either component, then first add verification part and then incorporate the tree construction mechanism, which results in our final WETAP, and we can investigate the gain by adding the component step by step:
>
> ### LLaMA-2-13B
> | Method     | MT-Bench M | MT-Bench Speed |  HumanEval M | HumanEval Speed |  GSM8K M | GSM8K Speed | Overall M | Overall Speed |
> |------------|------------:|---------------:|--------------:|-------------:|----------------:|---------------:|---------:|-------------:|
> | WETAP w/o tree and verification   | 6.14 | 31.74 | 6.92 | 35.98 | 6.38 | 32.96 | 6.48 | 33.56 |
> | WETAP w/o tree   | 6.94 | 33.20 (+1.46) | 7.88 | 39.77 (+3.79) | 6.96 | 35.44 (+2.48) | 7.26  | 36.14 (+2.58) |
> | WETAP  | 7.03 | 37.80 (+6.06) | 7.96 | 42.43 (+6.45) | 6.98 | 37.32 (+4.36) | 7.32 | 39.18 (+5.62) |
>
> The results above show that the standalone gain of deep-to-shallow verification is 2.58 and the marginal gain of dynamic tree construction based on the already present deep-to-shallow verification is 3.04. This demonstrates that dynamic tree construction provides a strong complementary effect. Therefore, it is not appropriate to judge the importance of it solely by its standalone gain; its contribution manifests when combined with deep-to-shallow verification, and the resulting improvement is comparable to (or even stronger than) that of deep-to-shallow verification.
>
> ```text
> The method seems somewhat sensitive to the choice of hyperparameters -- so it would be good to see a generalization study where the gamma and alpha are tuned on one dataset and used on another one.
> ```
>
> It is worth noting that hyperparameters $\alpha$ and $\gamma$ actually are **not so sensitive to datasets**. For the hyperparameter $\gamma$, we find that maintaining 1.2 can almost achieve similar inference speed in different datasets. For $\alpha$, the optimal one may change with datasets. So for the robustness of them, we **apply the best-performing hyperparameter settings of one dataset to others** to examine the changes in speed. The optimal ($\alpha$, $\gamma$) for MT-Bench is (0.3, 1.2), while (0.6, 1.2) for HumanEval and (0.6, 0.5) for GSM8K. The results are shown below:
>
> | ($\alpha$, $\gamma$) |      MT-Bench |     HumanEval |         GSM8K |
> | -------------- | ------------: | ------------: | ------------: |
> | (0.3, 1.2)           |     **37.80** | 41.25 (-2.8%) | 36.61 (-1.9%) |
> | (0.6, 1.2)           | 37.20 (-1.6%) |     **42.43** | 35.82 (-4.0%) |
> | (0.6, 0.5)           | 35.40 (-6.3%) | 41.55 (-2.1%) |     **37.32** |
>
> The results demonstrate that although using a sub-optimal combination, the speed will not degrade so much, which demonstrates that different combinations can achieve competitive performance and show relatively good robustness.
>
> In addition, it is expected that the optimal settings of these hyperparameters may vary across datasets, as different datasets naturally exhibit different categories and statistical characteristics.
>
> [1] Fast Inference from Transformers via Speculative Decoding.
> [2] Optimized multi-token joint decoding with auxiliary model for LLM inference.
> [3] Speculative Decoding with Big Little Decoder.
> [4] SWIFT: On-the-Fly Self-Speculative Decoding for LLM Inference Acceleration.

---

> > ### Comment · Reviewer_BxfY · 2025-11-26
> >
> > I thank the authors for their responses.
> >
> > 1. Thanks for clarifying that the verification procedure from MTAD does indeed remove the distributional guarantee of speculative decoding. However, the way the paper is presented currently does not make this point clear --- it is largely presented as an extension of speculative decoding, rather than that of MTAD. Hence, the implicit assumption that the reader will make is that we only need to care about the speedup. (The authors note that most SD papers just report generation length M and inference speedup, but this is precisely because we are guaranteed to preserve accuracy). As such the downstream results need to be a central focus of the paper.
> >
> > 2. For the downstream results, what is the "Vanilla" method and why is it so much worse than all the other ones. Based on the name, I would assume that "Vanilla" is just running the target model without the draft. It is not clear why this would be worse than approximating the target model with draft generations? Can you clarify where the accuracy gains are coming from?
> >
> > 3. What is "WETAP w/o tree and verification" --> how is the draft model used if you are not verifying the generations at all? Does this reduce to being standard speculative decoding, or just sampling from the target model itself? If yes to either of these questions, again it is not clear where does the speedup for this baseline come from?
> >
> > I appreciate the authors' efforts in addressing the weaknesses I pointed out in my original review, but the response raises more questions than it answers. This mirrors the presentation of the paper where baselines, methodological details are often presented without much context.

---

> > > ### Author Response · Authors · 2025-11-28
> > > **Response to Reviewer BxfY**
> > >
> > > Thanks for your response.  We are glad  for your appreciation for our clarification about the verification procedure. Below, we provide further clarifications to address your concerns. Please note that your original concerns are shown in framed boxes.
> > >
> > > ```text
> > > it is largely presented as an extension of speculative decoding, rather than that of MTAD. Hence, the implicit assumption that the reader will make is that we only need to care about the speedup. (The authors note that most SD papers just report generation length M and inference speedup, but this is precisely because we are guaranteed to preserve accuracy). As such the downstream results need to be a central focus of the paper.
> > > ```
> > >
> > > We sincerely appreciate your insightful comment. We agree that  our verification scheme diverges from the lossless setting and therefore requires downstream accuracy to be a primary evaluation focus, and we apologize for our lack of consideration and focusing primarily on speed in the initial submission. We appreciate you for pointing out this oversight and will clearly position our method as an extension of MTAD and emphasize downstream results accordingly.
> > >
> > >
> > >
> > > ```text
> > > For the downstream results, what is the "Vanilla" method and why is it so much worse than all the other ones. Based on the name, I would assume that "Vanilla" is just running the target model without the draft. It is not clear why this would be worse than approximating the target model with draft generations? Can you clarify where the accuracy gains are coming from?
> > > ```
> > >
> > > We sincerely apologize for not clarifying the definition of "Vanilla" in the previous submission. As you correctly assume, "Vanilla" refers to the single target model without the draft one. In addition, we appreciate your concern about the source of accuracy gains. We want to clarify that **the accuracy gains on MT-Bench come from the characteristics of this task and the assistance of the draft model** and want to clarify that **the accuracy gains on HumanEval come from the normal fluctuation in one or two more correct answers**.
> > >
> > > For **MT-Bench task**, it does not have the single ground-truth answer for each question, so the score by the judge model primarily depends on **the relevance of the final answer to the question as well as its grammaticality and logical coherence.** So the answer that exhibits the above properties more can receive higher scores, which is orthogonal to whether a method is lossless or not. **In addition**, compared to the vanilla target model, speculative decoding methods introduce an extra draft model which **enables a larger exploration space** by constructing a token tree, so they can achieve expression enhancement to get higher scores. And compared with lossless speculative decoding methods, lossy methods enable a more flexible exploration space with a bounded degree of relaxation and provide more diverse draft sequences to the target model to verify.
> > >
> > > For **HumanEval task**, a 2–3% accuracy fluctuation corresponds to only  a small number of questions being correct or not, so it is the normal fluctuation probably caused by stochastic sampling or other factors.
> > >
> > >
> > >
> > > ```text
> > > Can you clarify where the accuracy gains are coming from? What is "WETAP w/o tree and verification" --> how is the draft model used if you are not verifying the generations at all? Does this reduce to being standard speculative decoding, or just sampling from the target model itself? If yes to either of these questions, again it is not clear where does the speedup for this baseline come from?
> > > ```
> > >
> > > We sincerely apologize for our oversight to clarify the definition of **"WETAP w/o tree and verification".** Here "WETAP w/o tree and verification" refers to **WETAP without the dynamic tree construction and with the verification strategy the same as Speculative Decoding[1]**, so this method constructs the token tree with fixed width, prunes it only by probability, and verifies it token by token and layer by layer guaranteeing the distributional consistency. Compared to the vanilla target model itself, this method introduce an extra draft model and constructs a token tree to broaden the exploration space, so it achieves the speedup gain. In addition, **"WETAP w/o tree"** refers to **WETAP without dynamic tree construction but with the deep-to-shallow verification strategy**. So we start the experiments from the baseline without either component, then first add deep-to-shallow verification and then incorporate the dynamic tree construction mechanism, which results in our final WETAP to investigate the gains from each component.

---

> > > > ### Author Response · Authors · 2025-11-28
> > > > **Response to Reviewer BxfY**
> > > >
> > > > ```text
> > > > I appreciate the authors' efforts in addressing the weaknesses I pointed out in my original review, but the response raises more questions than it answers. This mirrors the presentation of the paper where baselines, methodological details are often presented without much context.
> > > > ```
> > > >
> > > > We are glad  for your appreciation for our efforts in addressing the weakness. And we also sincerely appreciate you for pointing out that several baselines and methodological components were not presented with sufficient context, which may lead to confusion for readers. We acknowledge and sincerely apologize for our oversight and shortcomings. We will revise our manuscript to provide clearer definitions, stronger motivation, and a more coherent presentation of the relationships among methods.
> > > >
> > > > ---
> > > >
> > > > We hope the above clarifications and additional experiments can comprehensively address your concerns. And we look forward to continuing the discussion with you. Thank you once again sincerely for the time and efforts you have dedicated to reviewing our submission.
> > > >
> > > >
> > > >
> > > > [1] Fast Inference from Transformers via Speculative Decoding.

---

### Author Response · Authors · 2025-12-03
**General Summarization - 1**

We sincerely thank all reviewers for their thoughtful and constructive feedbacks. Reviewers' comments have greatly helped us identify some shortcomings in our initial submitted manuscript, including clearer clarifications, more comprehensive experiments, and missing analyses. We also thank the AC's effort in carefully considering our responses. Below we summarize the key clarifications, additional experiments and in-depth discussions included in our revision.

---
## **Distributional Consistency (Lossless or Lossy)**

In general, if a speculative decoding method can guarantee the distributional consistency between the draft and target models, it is lossless, and vice versa. Here we want to discuss **if our WETAP is lossless or lossy when *T=0* and *T=1*.**

---

**• When *T=0***, our WETAP is **lossless** and **WETAP's final output sequence can match the target sequence.** We will explain our losslessness with a specific example below.

Assuming that we now have a draft token tree shown below:

```text
Draft Token Tree

O(prefix)
├─ A
│  └─ C
└─ B
   └─ D
```

The token tree has two layers where the prefix token *O* has two child tokens *A* and *B*, and they have one child token *C* and *D* respectively. Now we input the token tree into the target model and greedily sample one target child token based on each input draft token. It is worth noting that with greedy sampling, the probability of the final target child token is 1 and probabilities of all remaining tokens are 0:

```markdown
Target model greedily samples the target child token based on each draft token. For example, the target child token of token A is A' and its target probability p(A') must be 1.

O  -->  O' (p=1)
A  -->  A' (p=1)
B  -->  B' (p=1)
C  -->  C' (p=1)
D  -->  D' (p=1)
```

Now we can start the deep-to-shallow verification based on the draft token tree and target tokens. In specific, we **start from the deepest token *C*** and verify its corresponding sequence ***O -->A -->C***.  We verify this sequence by **the ratio** of its target cumulative probability p(A) · p(C) and its draft cumulative probability *q(A) · q(C)* (The probability of the prefix token *O* is 1). It is worth noting that if *p(A) · p(C)* is 1, the ratio $\frac{p(A) · p(C)}{q(A) · q(C)} = \frac{1}{q(A) · q(C)}$ will be more than 1, so this sequence will certainly be accepted. On the contrary, if *p(A) · p(C)* is 0, the ratio will be 0, meaning that this sequence will certainly be rejected. In the following we will clarify the verification:

```text
We start the verification of O -->A -->C from the deepest token C. We compare token C with its paren token A's target child token A'.

if C == A':
	p(C) = p(A') = 1  -->  continue the verification to the shallower token A
else:
	p(C) = 0  -->  p(A) · p(C) = 0, reject this sequence

Now we assume that token C = token A', so we continue the verification to the shallower token A by comparing it with token O'.

if A == O':
	p(A) = p(O') = 1  -->  now p(A) · p(C) = p(O') · p(A') = 1, accept this sequence
else:
	p(A) = 0  -->  p(A) · p(C) = 0, reject this sequence
```

As shown above, we will accept the draft sequence *O -->A -->C* **only when *A=O'* and *C=A'*** , where the **final accepted sequence is the same as the target sequence greedily sampled** by the target model. If any draft token in the sequence does not match the corresponding target one, **the cumulative target probability will be 0**, which leads to the rejection of this sequence. So the deep-to-shallow verification can still guarantee the losslessness when *T=0*.

In addition, we conduct experiments when *T=0* to investigate the inference speed. The results are shown below:

### LLaMA-2-13B

| Method    | MT-Bench M | MT-Bench Speedup | HumanEval M | HumanEval Speedup |  GSM8K M | GSM8K Speedup | Overall M | Overall Speed | Overall Speedup |
| --------- | ---------: | ---------------: | ----------: | ----------------: | -------: | ------------: | --------: | ------------: | --------------: |
| Vanilla   |       1.00 |            1.00× |        1.00 |             1.00× |     1.00 |         1.00× |      1.00 |         16.64 |           1.00× |
| SpecInfer |        3.8 |            1.36× |        4.85 |             1.68× |      4.1 |         1.47× |      4.25 |         24.96 |           1.50× |
| SpecExec  |       5.85 |            1.10× |        6.48 |             1.16× |     6.19 |         1.16× |      6.07 |         18.97 |           1.14× |
| MTAD      |       5.27 |            1.82× |        6.18 |             2.12× |      5.6 |         1.94× |      5.68 |         32.61 |           1.96× |
| **WETAP** |   **5.96** |        **1.97×** |    **6.75** |         **2.18×** | **6.40** |     **2.08×** |  **6.37** |     **34.55** |       **2.08×** |

Results above demonstrate that WETAP can **outperform** other methods when *T=0*.

---

> ### Author Response · Authors · 2025-12-03
> **General Summarization - 2**
>
> ---
> • **When *T=1***, our WETAP **can not guarantee the distributional consistency**, but **the degree of deviation can be controlled and bounded** by the overall quality of the sequence, and WETAP can **get higher speedup while maintaining competitive downstream performance.**
>
> Different from traditional Speculative Decoding, MTAD directly accepts a sequence based on its cumulative draft and target probability. This verification method may ignore little low-quality tokens in a sequence and accept it only considering the overall quality. Based on it, we discover that we can verify the tree from deep to shallow because we have already had draft and target probabilities, we can verify every token independently by its two cumulative probabilities instead of token by token and layer by layer, which destroys the distributional consistency.
>
> However, although our method destroys the consistency and allows minor deviation, **it will not accept sequences whose overall probability is clearly implausible or whose degradation is so substantial** caused by misleading shallow tokens. We accept a sequence based on its cumulative probability of all tokens, not based on the single probability of the final token in it. So our method relaxes the strictness of token-level verification, but it does not relax the global quality of the sequence.
>
> In addition, to demonstrate that our WETAP can achieve higher speedup while maintaining competitive downstream performance, we conduct experiments with Llama-2-13b on MT-Bench and HumanEval, using **vanilla target model, lossless methods** like SpecInfer and SpecExec, and **lossy methods** like MTAD, FSD, BiLD and our WETAP. We evaluate the downstream performance on MT-Bench by the score from deepseek-r1 as the judge model, and on HumanEval by pass@1. It is worth noting that we do not investigate the performance on GSM8K because the final performance of SD approaches is partly bounded by the vanilla target model Llama-2-13b which is not good at mathematical problems. Results are shown below:
>
> | Metric              | MT-Bench Speed (token/s) |   MT-Bench Score | HumanEval Speed (token/s) | HumanEval pass@1 |
> | ------------------- | -----------------------: | ---------------: | ------------------------: | ---------------: |
> | Vanilla Llama-2-13b |                    16.56 |            3.775 |                     16.56 |             18.7 |
> | SpecExec            |            17.55 (1.06×) |   5.862 (2.087↑) |             17.11 (1.03×) |      15.2 (3.5↓) |
> | SpecInfer           |            22.89 (1.38×) |     5.875 (2.1↑) |             26.21 (1.58×) |      15.2 (3.5↓) |
> | FSD                 |            20.57 (1.24×) |   5.888 (2.113↑) |             35.66 (2.15×) |      18.3 (0.4↓) |
> | MTAD                |            30.43 (1.84×) |    5.825 (2.05↑) |             38.00 (2.29×) |      16.5 (2.2↓) |
> | BiLD                |            35.74 (2.16×) |     4.275 (0.5↑) |             37.43 (2.26×) |      12.2 (6.5↓) |
> | **WETAP**           |        **37.80 (2.28×)** | **5.975 (2.2↑)** |         **42.43 (2.56×)** |  **18.9 (0.2↑)** |
>
> Compared to other **lossy and lossless methods**, our WETAP **outperforms them** in terms of both inference speedup and downstream performance. Compared to **vanilla Llama-2-13b target model**, WETAP **can even outperform it** on the downstream performance.
>
> 1. **For the dialogue task MT-Bench**, it does not have the single ground-truth answer, so the score primarily depends on the relevance of the final answer to the question as well as its grammaticality and logical coherence. So our WETAP can outperform vanilla target model because it includes an extra draft model to enhance expression and enables a larger and more flexible exploration space to generate more comprehensive answers by an appropriately loose verification.
> 2. **For the coding task HumanEval**, WETAP outperforms vanilla target model due to the normal fluctuations in one or two more correct answers.
> 3. **To sum up,** the appropriately loose verification can improve the speed while incurring only limited quality degradation, and in some cases may even lead to quality gains due to more dynamic answers.

---

> ### Author Response · Authors · 2025-12-03
> **General Summarization - 3**
>
> ## **Importance of the Two Components: Dynamic Tree Construction and Deep-to-shallow Verification**
>
> It is worth noting that our manuscript only conducts the ablation experiments on entropy-width construction and tree pruning method separately, but does not regard these two components above as **an integrated dynamic tree construction** and does not conduct ablation experiments on dynamic tree construction and deep-to-shallow verification. So it may not be clear to investigate the importance of  two components above in WETAP.
>
> In the following, we divide WETAP into two components and conduct **ablation experiments on dynamic tree construction and deep-to-shallow verification.** It is worth noting that ***baseline*** refers to WETAP without dynamic tree construction and without deep-to-shallow verification, ***baseline + d2sverification*** refers to WETAP with deep-to-shallow verification but without dynamic tree construction, ***baseline + tree*** refers to WETAP with dynamic tree construction but without deep-to-shallow verification, ***WETAP***  refers to the full WETAP with two components above.
>
> ### LLaMA-2-13B
>
> |           Metric           | MT-Bench M |               MT-Bench Speed                | HumanEval M |               HumanEval Speed               | GSM8K M |                 GSM8K Speed                 | Overall M |               Overall Speed                |
> | :------------------------: | :--------: | :-----------------------------------------: | ----------- | :-----------------------------------------: | :-----: | :-----------------------------------------: | :-------: | :----------------------------------------: |
> |          baseline          |    6.14    |                    31.74                    | 6.92        |                    35.98                    |  6.38   |                    32.96                    |   6.48    |                   33.56                    |
> | baseline + d2sverification |    6.94    |               33.20  (+1.46)                | 7.88        |                39.77 (+3.79)                |  6.96   |               35.44  (+2.48)                |   7.26    |               36.14  (+2.58)               |
> |      baseline + tree       |    6.14    |               32.91  (+1.17)                | 7.83        |               40.50  (+4.52)                |  6.49   |               34.95  (+1.99)                |   6.82    |               36.12  (+2.56)               |
> |           WETAP            |    7.03    | 37.80  (+6.06, +4.60 vs d2s, +4.89 vs tree) | 7.96        | 42.43  (+6.45, +2.66 vs d2s, +1.93 vs tree) |  6.98   | 37.32  (+4.36, +1.88 vs d2s, +2.37 vs tree) |   7.32    | 39.18 (+5.62, +3.04 vs d2s, +3.06 vs tree) |
>
>
> Results above show that:
>
> 1. **For generation length $M$,** deep-to-shallow verification yields a  greater improvement on $M$ compared with dynamic tree construction. This is because they **contribute in different ways.** Deep-to-shallow verification focuses on a loose verification standards to improve $M$, whereas dynamic tree construction focuses on accurately allocating and retaining the construction resources in the token tree. Therefore the dynamic tree construction can improve the speed through reducing unnecessary computation, while it may not effectively enhance $M$.
>
> 2. **For standalone speed gain,** the standalone speed gain of deep-to-shallow verification is 2.58, and 2.56 for the standalone dynamic tree construction. Although the standalone gain of deep-to-shallow verification is slightly larger than that of dynamic tree construction, this does not directly imply that dynamic tree construction is less important, because they contribute in fundamentally different ways which can be seen from the generated length $M$. More importantly, no matter which component is added individually to the baseline, they **both outperform** most of SOTA methods, highlighting **the importance and effectiveness of each one** in contributing to the overall speedup.
>
> 3. **For marginal speed gain**, it is obtained by comparing WETAP to each single-component variant to measure each component's improvement when the other is already present. The marginal gain of dynamic tree construction (*baseline + d2sverification → WETAP*) 3.04 and deep-to-shallow verification (*baseline + tree → WETAP*) 3.06 are **both larger than their standalone gain**, 2.56 and 2.58 respectively. This suggests that the contribution of each single component is not limited to its standalone behavior **but emerges more prominently in combination with the other one, highlighting the complementarity and synergy between them.**
>
> 4. **To sum up,** instead of ranking these components by their standalone speed gain, we view them as two mutually reinforcing components whose **combination (full WETAP) is what ultimately matters for the final inference speed.**

---

> ### Author Response · Authors · 2025-12-03
> **General Summarization - 4**
>
> ## **Comparison and Combination with EAGLE**
>
> ### **• Comparison with EAGLE:**
>
> We want to clarify that comparison between our training-free WETAP and training-based methods like EAGLE is not absolutely fair:
>
> ---
>
> **(1) Computational and time overhead of training-based methods are unacceptable and unfeasible in some cases.**
>
> Training-based methods like EAGLE and Medusa all require additional training to get higher speedup, which require substantial time and overhead. For example: EAGLE requires **1–2 days of training with 8 RTX 3090 GPUs** for LLaMA-33B or **up to 2 days on 4 A100 (40G) GPUs** for LLaMA2-Chat-70B. Although EAGLE has released many checkpoints, users still need to spend substantial time and computational resources retraining a new drafter if their target model is not supported or if they wish to leverage their own custom datasets to train the drafter.
>
> ---
>
> **(2) The high speedup achieved by training-based methods can not diminish the research value of plug-and-play approaches like WETAP.**
>
> Plug-and-play SD approaches are usually **free of training and model-agnostic.** More importantly, they can be **used and achieve inference speedup directly** in no require of additional computational and time overhead. Compared to training-based methods, they have broader applicability and are highly practical in real-world scenarios that demand high efficiency and easy integration.
>
> ---
>
> ### **• Combination with EAGLE:**
>
> Although it is not absolutely fair to compare training-free WETAP with training-based EAGLE, it is a promising research direction to combine both of them:
>
> 1. As mentioned above, **WETAP is plug-and-play** so it can be easily combined with EAGLE without any additional adjustments.
> 2. WETAP focuses on plug-and-play acceleration about the dynamic tree construction and deep-to-shallow verification, while EAGLE pursues improvement through learned auxiliary models. So these two research directions **are orthogonal and can in principle be combined to leverage their respective strengths.**
>
> The combination between them **precisely reflects WETAP's plug-and-play nature.**
>
> To investigate the feasibility, we conduct experiments on **vanilla llama-2-13b model, EAGLE and EAGLE + WETAP** when *T=0.* The results are shown below:
>
> ### LLaMA-2-13B
>
> | Method              | MT-Bench Speed | MT-Bench Speedup | HumanEval Speed | HumanEval Speedup | GSM8K Speed | GSM8K Speedup | Overall Speed | Overall Speedup |
> | ------------------- | -------------: | ---------------: | --------------: | ----------------: | ----------: | ------------: | ------------: | --------------: |
> | Vanilla Llama-2-13b |          16.56 |            1.00× |           16.56 |             1.00× |       16.81 |         1.00× |         16.64 |           1.00× |
> | EAGLE               |          29.26 |            1.77× |           30.91 |             1.87× |       33.13 |         1.97× |         31.10 |           1.87× |
> | **EAGLE + WETAP**   |      **32.32** |        **2.25×** |       **35.87** |         **2.17×** |   **36.69** |     **2.18×** |     **34.96** |       **2.10×** |
>
> The results above demonstrate that **the combination of WETAP and EAGLE can further improve the inference speedup** compared to EAGLE itself, highlighting the feasibility and advantages of combining them.

---

> ### Author Response · Authors · 2025-12-03
> **General Summarization - 5**
>
> ## **Validity of Deep-to-shallow Verification Strategy**
>
> Whatever the traversal direction of the token tree, we still need to input all tokens into the target model to get respective target probability. So our opposite traversal direction does not influence the forward pass of the target model but **influences the verification progress in the token tree.**
>
> The opposite traversal direction can prioritize exploring the longest candidate sequences, enabling the earliest discovery of a feasible sequence, and prioritize eliminating the most unstable candidates, rapidly shrinking the search space.
>
> In addition, the accepted misleading shallow tokens through deep-to-shallow verification **may not necessarily cause a persistent impact on subsequent token generation**, also demonstrating its validity. Below are two reasons:
>
> 1. Our verification is based on the cumulative probability of all tokens in a sequence, so actually it  will not accept sequences that are severely misled by misleading shallow tokens. Only when the deviation caused by shallow misleading tokens is **small enough not to cause a substantial drop in the quality** of the whole sequence, it will be accepted.
>
> 2. When the model generates tokens in an autoregressive way, it puts its attention on all previous tokens to generate subsequent tokens, so even when shallow tokens deviate slightly, the subsequent token distribution may can still naturally steer back toward high-probability semantic region.
> ---
> ## **Sensitivity of Hyperparameters**
>
> It is common for deep learning models and methods to exhibit a certain degree of hyperparameter sensitivity across datasets, as different datasets naturally exhibit diverse categories and statistical characteristics. However, in our WETAP, the **hyperparameters $\alpha$ and $\gamma$ are not particularly sensitive to datasets** in practice.
>
> To assess robustness of them more concretely, we conduct a **transfer experiment: we apply the best-performing hyperparameter setting of one dataset to others** to examine the changes in inference speed. It is worth noting that the optimal ($\alpha$, $\gamma$) for MT-Bench is (0.3, 1.2), while (0.6, 1.2) for HumanEval and (0.6, 0.5) for GSM8K. The results are shown below:
>
> | ($\alpha$, $\gamma$) |      MT-Bench |     HumanEval |         GSM8K |
> | -------------------- | ------------: | ------------: | ------------: |
> | (0.3, 1.2)           |     **37.80** | 41.25 (-2.8%) | 36.61 (-1.9%) |
> | (0.6, 1.2)           | 37.20 (-1.6%) |     **42.43** | 35.82 (-4.0%) |
> | (0.6, 0.5)           | 35.40 (-6.3%) | 41.55 (-2.1%) |     **37.32** |
>
> The results show that although using a sub-optimal hyperparameter settings, WETAP **exhibits strong stability**: five out of six transferred settings show less than 4% degradation, and even the worst case remains within a modest 7% drop. More importantly, WETAP using the sub-optimal  hyperparameter settings can still **outperform** other methods.
>
> Taken together, these results indicate that WETAP can maintain competitive performance across different hyperparameter settings and demonstrate relatively strong robustness.
>
> ---
> ## **Interpretation of Perplexity**
>
> As mentioned above, our WETAP can guarantee losslessness when *T=0* while are lossy when *T=1*. So we want to use **perplexity to serve as a proxy for the degree of lossiness when lossy.**
>
> Lossiness in our WETAP mostly comes from the loose deep-to-shallow verification based on sequence level, so some tokens in an accepted sequence may deviate from the corresponding target distribution. **The more they deviate from the target distribution, the smaller their corresponding target probabilities will be, which will increase the final perplexity.** So perplexity can measure the degree of this deviation and act as a proxy for the degree of lossiness.
>
> However, **perplexity can only serve as an indirect indicator** because it measures distributional deviation rather than the verification behavior itself. Lossiness essentially arises from the loose verification allowing deviated tokens to be mistakenly accepted, but perplexity can not directly represent these decisions. So we can use the **downstream performance on specific datasets** to investigate the impact of lossiness on the final quality more directly and effectively, which have been shown above.

---

> > ### Author Response · Authors · 2025-12-03
> > **General Summarization - 6**
> >
> > ## **Per-step Latency in Draft Inference**
> >
> > The additional operations introduced to the draft stage include calculating each layer's entropy and width, and additional operations brought by the new tree pruning method. **The respective extra latency brought by them is in practice negligible to the forward time of the draft model.**
> >
> > To investigate their impacts, we conduct two experiments with TinyLLama-1.1B as the draft model. One is **our method WETAP** including two extra operations mentioned above and the other one uses the **traditional fixed-width tree construction and probability-pruning method.** Then we investigate the respective time of width calculation, draft forward pass and tree pruning, and investigate the total time needed in draft stage. The results are shown below (**unit is second**):
> >
> > |                 | Width&Entropy Calculation | Forward Pass | Tree Pruning | Total Draft |
> > | --------------- | ------------------------: | -----------: | -----------: | ----------: |
> > | **WETAP**       |                     0.001 |       0.1183 |       0.0083 |      0.1365 |
> > | **Traditional** |                         0 |       0.1145 |       0.0072 |      0.1314 |
> >
> > • **For the additional width and entropy calculation**, it introduces almost no additional latency. In addition, theoretically width-entropy operation introduces just ***5T+1* FLOPS for a layer with *T* tokens.** Because *T* falls in the range from 16 to 128, so it is negligible compared to every matrix calculation in one model forward pass.
> >
> > • **For the tree pruning**, WETAP's new tree pruning method just takes 0.0011 seconds more than the traditional one, which is also negligible compared to the forward time by the draft model.
> >
> > ---
> > ## **Complexity of Tree Pruning Method**
> >
> > Regarding the complexity of pruning the token tree, we assume that the token tree has $N$ tokens and $L$ layers, and we want to prune it to $W$ tokens at last:
> >
> > 1. We prune the tree to $W$ tokens first based on probability and length, whose complexity is $O(N \log N)$.
> > 2. Then we need to supplement omitted parent tokens into the token tree, whose complexity is $O(N)$.
> > 3. Now the number of tokens in the tree may be more than *W*, so we need to prune it again by dropping leaf tokens from shallow to deep layers, whose complexity is $O(N)$.
> > 4. If the number now is still more than $N$, we need to prune it at last based on probability, whose complexity is $O(N \log N)$.
> >
> > In summary, **the total complexity is $O(N \log N)$, which is in the same complexity class as the traditional pruning method** just based on probability. So our pruning method can retain more potential tokens **without increasing the asymptotic computational cost.**
> >
> > ---
> > We sincerely hope these additional results and analyses could comprehensively address the reviewers’ concerns, demonstrate the robustness as well as efficiency achieved by WETAP and enhance the overall quality of our manuscript. We also hope this summary helps the AC in evaluating the submission.

---

### Meta-Review · Area_Chair_7kVp · 2026-01-13

**Summary:**

This paper proposes WETAP, a training-free speculative decoding method with dynamic token tree construction based on entropy and deep-to-shallow verification. Strengths include the intuitive idea of entropy-based width allocation, strong speedup numbers, and the ability to combine with training-based methods like EAGLE. However, there are significant weaknesses: (1) The deep-to-shallow verification strategy breaks distributional consistency guarantees of speculative decoding, making this a lossy method, but the paper was initially presented as an extension of standard SD rather than lossy approaches. (2) The theoretical justification for why accepting deep tokens implies parent tokens are acceptable is weak, shallow misleading tokens could steer outputs into problematic regions. (3) The presentation lacks clarity on baselines and methodological details. (4) Hyperparameters require task-specific tuning. (5) Missing comparison with trained draft methods (EAGLE, Medusa).

**Reviewer Concerns:**

Addressed: (1) Downstream performance experiments were added showing WETAP achieves competitive MT-Bench scores and HumanEval pass@1. (2) T=0 experiments were provided showing losslessness under greedy decoding. (3) Hyperparameter transfer experiments showed <7% degradation across datasets. (4) EAGLE+WETAP combination experiments showed additional speedup gains. (5) Latency breakdown confirmed negligible overhead from entropy/pruning operations. Outstanding: (1) The core concern about deep-to-shallow verification validity remains unresolved: why does acceptance of a deep token imply parent tokens are acceptable? The cumulative probability argument relaxes verification but doesn't address shallow misleading tokens. (2) The paper's framing as an extension of SD rather than lossy speculative decoding caused confusion about losslessness guarantees. (3) One reviewer explicitly stated "the response raises more questions than it answers" regarding presentation clarity.

**Reviewer Scores:**

Reviewer BxfY (2): Would likely maintain score. Explicitly stated "the response raises more questions than it answers" and remained concerned about presentation clarity and the lossy nature of the method.

Reviewer Nves (6): Would likely maintain score. Their concerns about pruning complexity and verification direction were partially addressed.

Reviewer YL9S (4): Would likely maintain score. Core concern about deep-to-shallow verification validity remained unresolved despite multiple rounds of discussion. Explicitly stated concerns were not fully addressed.

Reviewer G78s (6): Would likely maintain score. Concerns about computational overhead and hyperparameter sensitivity were partially addressed with experiments.

---

### Decision · Program_Chairs · 2026-01-26

Reject